# Palmitoylation by ZDHHC4 inhibits TRPV1-mediated nociception

Youjing Zhang [ID][1,4], Mengyu Zhang[1,4], Cheng Tang [ID][2,4], Junyan Hu[1], Xufeng Cheng[1], Yang Li[1], Zefeng Chen[1], Yuan Yin[2], Chang Xie[1], Dongdong Li [ID][3✉] & Jing Yao [ID][1✉]

## Abstract

**Transient receptor potential vanilloid 1 (TRPV1) is a capsaicin-sensitive ion channel implicated in pain sensation. While TRPV1 potentiation in hyperalgesia development has been extensively investigated, its functional decline during pain relief remains largely unexplored. Here, by molecular, electrophysiological and in vivo evidence, we reveal that S-palmitoylation fine-tunes TRPV1 function by promoting its degradation via the lysosome pathway thereby facilitating inflammatory pain relief. The palmitoyl acyltransferase ZDHHC4 is identified to physically interact with TRPV1 and to catalyze S-palmitoylation at the cysteine residues C157, C362, C390, and C715 of the channel. Furthermore, we show that TRPV1 palmitoylation is counterbalanced by the depalmitoylase acyl-protein thioesterase 1 (APT1), thereby reinstating pain sensation. These findings provide important mechanistic insights into the relief phase of inflammatory pain.**

Keywords TRPV1; ZDHHC4; Palmitoylation; Inflammatory Pain; Protein Degradation
Subject Categories Membranes & Trafficking; Neuroscience; Post-translational Modifications & Proteolysis

## Introduction

TRPV1 is a cation channel primarily expressed in small to medium-sized nociceptive neurons of the dorsal root ganglia (Caterina et al, 1997; Thomas et al, 2011; Tominaga et al, 1998). TRPV1 mediates cation influx into the neuron to initiate and propagate action potentials in response to noxious stimuli (Jara-Oseguera et al, 2008), thereby contributing to pain sensation. Targeting TRPV1 holds the potential to develop novel analgesics to counter the misuse of addictive pain killers, such as the worldwide opioid crisis in pain treatment (Simpson et al, 2023).

The experience of acute pain serves as a physiological response to adverse stimuli, acting as a protective mechanism to forewarn of further harm. Within the context of inflammation, the inflammatory factors sensitize ion channels involved in pain signal transduction, including TRPV1. Multiple molecular mechanisms have been identified for TRPV1 sensitization during pain development. A key regulatory aspect of TRPV1 is its post-translational modification. The cytoplasmic domain of TRPV1 undergoes phosphorylation by protein kinase C (PKC) and protein kinase A (PKA) (Bhave et al, 2002; Numazaki et al, 2002). PKC phosphorylation at Ser502 and Ser800 is crucial for potentiating TRPV1 activation by phorbol ester, ATP, and heat (Numazaki et al, 2002), whereas PKA-mediated phosphorylation at Ser116 is essential for inhibiting capsaicin-induced desensitization (Bhave et al, 2002). Another protein kinase, calmodulin-dependent kinase II (CaMKII), mediates the phosphorylation of TRPV1 at sites Ser502 and Thr704, affecting its binding to the prototypical ligand, capsaicin (Jung et al, 2004). In addition, Cdk5, phosphorylates Thr407 on TRPV1, altering capsaicin sensitivity and heat responsiveness (Jendryke et al, 2016). Furthermore, SUMOylation of TRPV1 at the Lys822 site exacerbates heat hyperalgesia during inflammation (Wang et al, 2018). Phosphorylation also plays a role in regulating the forward trafficking of TRPV1. Src kinase, activated by signaling, has been reported to phosphorylate the Tyr200 site of TRPV1, inducing its translocation to the cell surface from the vesicular pool (Zhang et al, 2005).

Beyond post-translational modification, protein-protein interaction also regulates TRPV1 sensation in chronic pain. Multiple regulatory proteins, such as PI3K, GABAA receptor-associated proteins, voltage-gated potassium channel auxiliary subunit Kvβ1, KChIP3, and Whirlin, have been shown to facilitate the translocation of TRPV1 to the plasma membrane during hyperalgesia (Ciardo et al, 2016; Laínez et al, 2010; Stein et al, 2006; Tian et al, 2018; Wang et al, 2020). Furthermore, the intracellular recycling of TRPV1 affects its surface protein levels and tachyphylaxis reaction, through distinct pathways mediated by synaptotagmin 1 and 7, respectively (Tian et al, 2019). Lipids also contribute to modulate TRPV1 function. The depletion of PIP2 coincides with the activation of TRPV1, and its replenishment in the membrane determines the recovery of the channel from desensitization (Liu et al, 2005). Transcription factors, on the other hand, regulate TRPV1 by interfering with its de novo synthesis. ZBTB20, a crucial regulator of nociception and pain sensation, indirectly upregulates TRPV1 protein levels (Ren et al, 2014). Additionally, transcription

[1]State Key Laboratory of Virology, TaiKang Center for Life and Medical Sciences, College of Life Sciences, Frontier Science Center for Immunology and Metabolism, Hubei Key Laboratory of Cell Homeostasis, Wuhan University, Wuhan, Hubei 430072, China. [2]The National and Local Joint Engineering Laboratory of Animal Peptide Drug Development, College of Life Sciences, Hunan Normal University, Changsha, Hunan 410081, China. [3]Sorbonne Université - CNRS - INSERM, Institut de Biologie Paris Seine, Neuroscience Paris Seine, Paris 75005, France. [4]These authors contributed equally: Youjing Zhang, Mengyu Zhang, Cheng Tang. ✉E-mail: dongdong.li@inserm.fr; jyao@whu.edu.cn

factors Sp1 and Sp4 enhance TRPV1 expression by binding to the candidate GC-box site within the endogenous TRPV1 P2-promoter region (Chu et al, 2011). These findings have advanced the molecular understanding of TRPV1 sensation in pain development. Meanwhile, targeting pain development has been found to be limited in obtaining effective clinical analgesia. Despite its protective role in alerting noxious environments, prolonged and intense pain can lead to physical and mental disability (Loeser and Melzack, 1999; Macchia and Oswald, 2021). During animal nociceptive reaction, therefore, pain sensation needs to be timely relieved to avoid chronic intractability and long-term perturbations in sensory and psychological homeostasis. Pinpointing the intrinsic pathways regulating TRPV1 functional declining during the pain relief phase will help to understand how the body re-equilibrates the sensory processing capability, while providing molecular targets to temporally control pain sensation.

In the present study, we have elucidated the mechanisms that regulate TRPV1 protein levels in mouse DRG neurons during the pain relief phase. Through mesoscale mRNA screening, we identified palmitoyl transferase ZDHHC4 directly interacting with TRPV1. Our biochemical, electrophysiological, and in vivo behavioral results demonstrated that ZDHHC4 mediates the palmitoylation of TRPV1 during the inflammatory pain relief process, promoting TRPV1 degradation via the lysosomal pathway. This process was controlled by the depalmitoylase APT1. Furthermore, we have pinpointed the four ZDHHC4 palmitoylation sites within TRPV1, namely Cys157, Cys362, Cys390, and Cys715. These findings thus shed light on TRPV1 functional regulation during the recovery of inflammatory pain, which may serve as potential therapeutic targets for pain management interventions.

# Results

## Pain relief is accompanied by TRPV1 downregulation and enhanced TRPV1 palmitoylation

To interrogate the role of TRPV1 in pain sensation and relief, we first analyzed the pain time-course in a carrageenan-induced mouse inflammatory pain model, and monitored TRPV1 protein levels in the nociceptive dorsal root ganglion (DRG) neurons. As expected, the wild-type mice developed evident pain behavior in both mechanical allodynia and thermal hyperalgesia tests post-carrageenan application, as illustrated by remarkably decreased paw withdrawal threshold and reduced paw withdrawal latency, respectively (Fig. 1A,B). In contrast, $Trpv1^{-/-}$ mice were insensitive to carrageenan without developing nociceptive behavior, similar to the group receiving the control saline (Fig. 1A,B). $Trpv1^{-/-}$ mice exhibited enhanced tolerance to nociceptive thermal stimuli than the wild-type mice as well (Fig. 1B). These findings comply with previous studies arguing for the pivotal roles of TRPV1 in pain perception. The pain intensity peaked at 12 h post carrageenan administration, and was reduced thereafter, reflecting the pain relief phase. We observed that the thermal hyperalgesia disappeared much faster (within 24 h) than mechanical allodynia (Fig. 1A,B). Interestingly, there was a close correlation between TRPV1 protein levels and pain intensity, which both remained constant in the saline group but simultaneously increased to peak at 12 h and decreased to baseline thereafter in the carrageenan group (Fig. 1A–D), suggesting an involvement of TRPV1 downregulation in pain relief.

To screen at biochemical levels the principal pathways underlying TRPV1 degradation, we utilized the ND7/23 neuroblastoma cell line, a hybrid of mouse neuroblastoma and rat DRG neurons with endogenously expressed TRPV1 (Wood et al, 1990). By inhibiting the lysosomal, proteasomal, endosomal, and autophagic protein degradation pathways with their specific inhibitors, Bafilomycin A1 (BafA1) (Drose and Altendorf, 1997), Z-Leu-Leu-Leu-CHO (MG132) (Guo and Peng, 2013), or 3-methyladenine (3MA) (Rusilowicz-Jones et al, 2022; Wu et al, 2010), respectively, we observed that only blocking the lysosomal pathway slowed down TRPV1 degradation, resulting in time-dependent protein accumulation (Fig. 1E,F). In the meantime, we excluded the ubiquitin-proteasome pathways for TRPV1 degradation (Appendix Fig. S1). These data suggest that TRPV1 degradation is mainly mediated by the lysosomal pathway. We then examined the potential involvement of post-translation modifications in TRPV1 degradation, among which palmitoylation is a critical post-translational modification controlling protein stability (Linder and Deschenes, 2007), lysosomal sorting and degradation (Bagh et al, 2017; Canto and Trejo, 2013). We thus conducted the acyl-biotin exchange (ABE) assay (Brigidi and Bamji, 2013) assessing the S-palmitoylation of TRPV1 in DRG neurons from the pain model mice. Indeed, during the pain relief phase post carrageenan application, the intensity of TRPV1 palmitoylation consistently increased between 12–24 h, coinciding with the decrease in TRPV1 protein levels (Fig. 1G,H). Collectively, these findings suggest that TRPV1 palmitoylation contributes to its lysosomal degradation, promoting inflammatory pain resolution.

## The palmitoyl transferase ZDHHC4 physically interacts with TRPV1

The zinc finger DHHC domain-containing (ZDHHC) proteins mediate cellular S-palmitoylation reactions and 23 different ZDHHC palmitoylase are present in mammals. To determine which palmitoylase mediates TRPV1 palmitoylation, we first examined the mRNA transcription of all 23 ZDHHCs in DRG neurons of the pain model mice, at three specific time points corresponding to significant changes in TRPV1 protein levels (0, 12, and 24 h). As a result, the ZDHHC2, ZDHHC4, ZDHHC12, ZDHHC15, ZDHHC17, and ZDHHC18 mRNA levels at 12 and/or 24 h were remarkably changed when compared to their respective baseline at 0 h (Fig. 2A), implying their potential roles in TRPV1 palmitoylation. We thus coexpressed these ZDHHCs (GFP tagged) with TRPV1 (Flag-tagged) in HEK293T cells and tested their possible interactions by co-immunoprecipitation (CO-IP) assays. It was found that the palmitoylase ZDHHC4, but not the others, interacted with TRPV1 and considerably downregulated its protein levels (Fig. 2B). Moreover, the ZDHHC4 mRNA levels progressively increased over time (Fig. 2A), which is in line with the consistently enhanced palmitoylation of TRPV1 (Fig. 1G,H). This interaction was confirmed in situ by clear colocalization of mCherry-tagged TRPV1 and GFP-tagged ZDHHC4 on the plasma membrane (PM) in HEK293T cells imaged with confocal microscopy (Appendix Fig. S2A). Additionally, we used bimolecular fluorescence complementation (BiFC) imaging to validate the physical interaction between ZDHHC4 and TRPV1 in cells (Hu et al, 2021). We observed the reconstituted green fluorescence on cell PM brought by the interaction between CsfGFP-tagged TRPV1

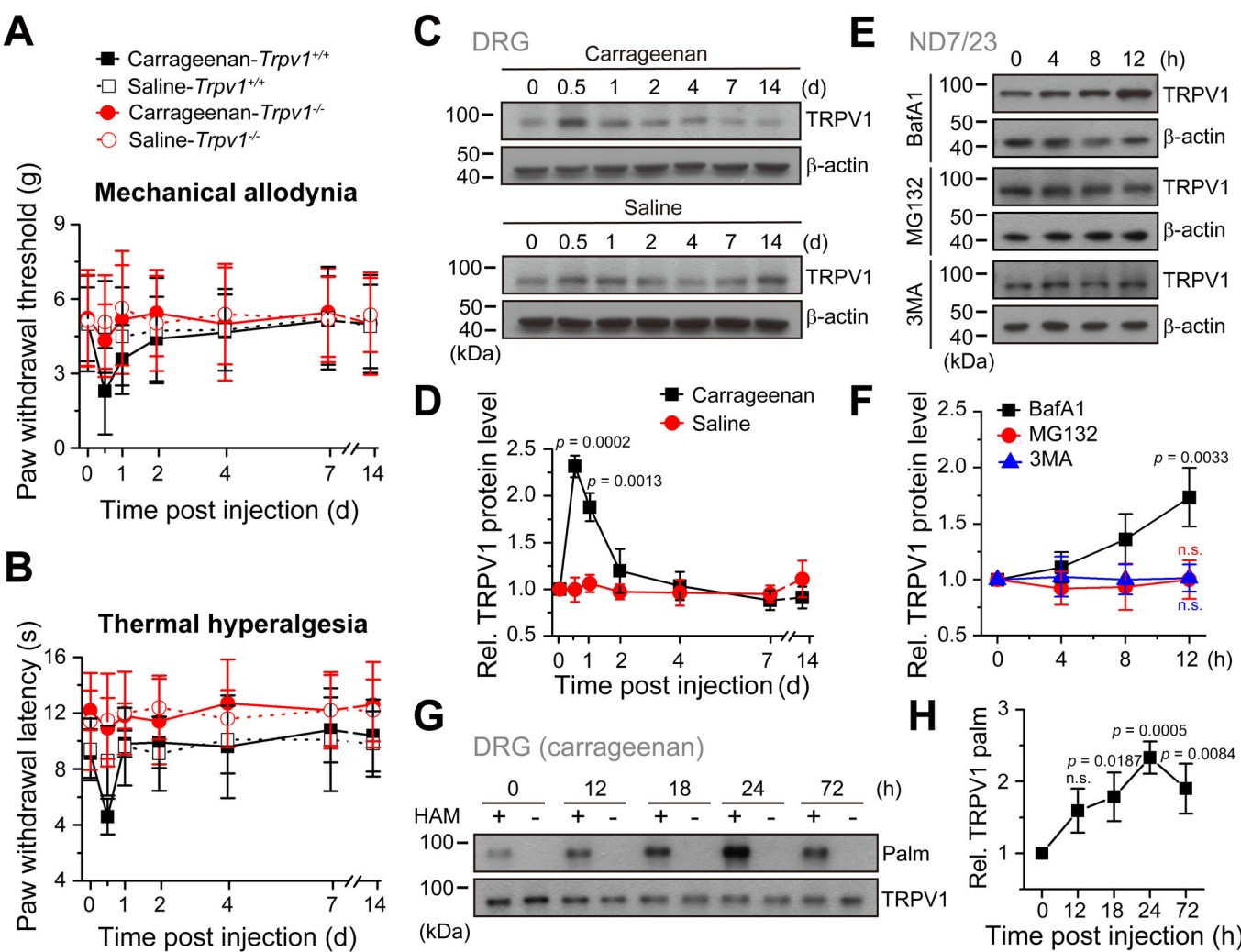

**Figure 1. Downregulated protein levels and upregulated palmitoylation of TRPV1 during pain relief.**

(A, B) Carrageenan (2%, 20 µl) or saline (20 µl) was administered by intraplantar injection into the right hindpaw of wild-type or *Trpv1⁻/⁻* mice (*n* = 10 mice for each group). The radiant heat test and Von Frey filament assay shows carrageenan-induced mechanical allodynia (A) and thermal hyperalgesia (B) in wild-type but not TRPV1 knock-out (*Trpv1⁻/⁻*) mice, mice injected with saline served as the negative control. (C) Immunoblotting depicting TRPV1 protein levels in DRG neurons (L4–L6) of mice following intraplantar injection of 20 µL carrageenan (upper panel) in the right hindpaw (2%, w/v, in saline) or saline (lower panel) in the left hindpaw (20 µL) over different time intervals; β-actin used as the loading control. One representative blot of three replicate experiments is shown. (D) Normalized TRPV1 protein level in (C) through gray intensity analysis, with statistical significance assessed using unpaired Student's *t*-test (carrageenan vs. saline at the same time point post injection) (*n* = 3, *n* represents biological replicates). (E) Immunoblotting illustrating TRPV1 protein levels in ND7/23 cells at various time points after treatment with inhibitors of proteasomal (MG132), autophagic (3MA), and lysosomal (BafA1) protein degradation pathways, β-actin used as the loading control. One representative blot of three replicate experiments is shown. (F) The normalized TRPV1 protein level in (E) through gray intensity analysis, with statistical significance assessed using one-way ANOVA with post hoc Dunnett analysis, TRPV1 protein level at 0 h served as the control in each group (*n* = 3, *n* represents biological replicates). (G) Detection of TRPV1 palmitoylation in mouse DRG neurons at different time points post intraplantar carrageenan injection using the acyl-biotinyl exchange (ABE) assay, with TRPV1 per se serv as the loading control. One representative blot of three replicate experiments is shown. (H) The palmitoylation levels of TRPV1 in (G) was quantified using a gray intensity analysis, with statistical significance assessed using one-way ANOVA with post hoc Dunnett analysis, TRPV1 palmitoylation level at 0 h served as the control (*n* = 3, *n* represents biological replicates). Data information: In (A, B, D, F, H), data were presented as mean ± SD. The *p* values are shown in the figure. n.s., not significant. Source data are available online for this figure.

and NsfGFP-tagged ZDHHC4, while no reconstituted fluorescence observed when expressing either TRPV1-Csf or ZDHHC4-Nsf with their respective GFP control vectors (GFP-Nsf or GFP-Csf) (Appendix Fig. S2B). To probe the interacting domain in TRPV1 with ZDHHC4, we constructed three FLAG-tagged TRPV1 truncates and analyzed their co-immunoprecipitation with ZDHHC4. As illustrated in Fig. 2C, we observed that ZDHHC4 strongly precipitated with the TRPV1 N-terminus (aa

1–432) as well as the C-terminus (aa 687–839), while the TRPV1 transmembrane domain is unlikely involved. The interaction between ZDHHC4 and TRPV1 was further verified in native DRG neurons (Fig. 2D).

Previous studies have reported that TRPV1 activation leads to the activation of JAK2-STAT3 signaling (Yang et al, 2019), and the phosphorylated STAT3 promotes ZDHHC4 transcription (Zhao et al, 2022). We therefore examined the phosphorylation levels of

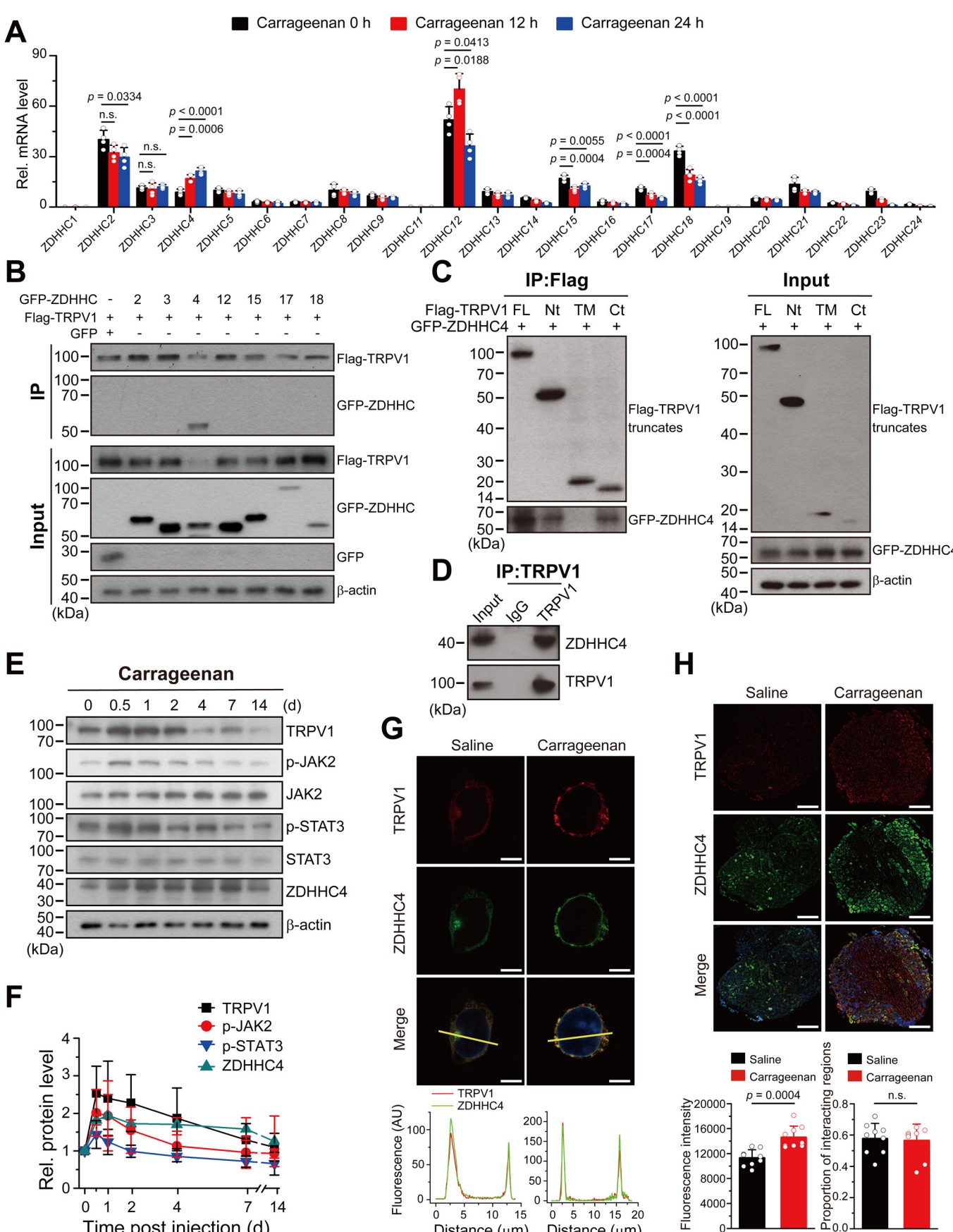

**Figure 2. The palmitoyl transferase ZDHHC4 physically interacts with TRPV1.**

(A) Real-time qPCR was employed to assess the relative mRNA level of various ZDHHCs in mouse DRG at three time points (0, 12, and 24 h) following intraplantar carrageenan injection, with β-actin serving as the internal control. The statistical differences between the groups were assessed using one-way ANOVA with post hoc Dunnett analysis. The relative mRNA level of each ZDHHC at 0 h was employed as the control. Each data point represents a separate biological replicate ($n = 4$, $n$ represents biological replicates). (B) Co-immunoprecipitation, utilizing the anti-flag antibody, investigated the interaction between FLAG-tagged TRPV1 (FLAG-TRPV1) and seven different GFP-tagged ZDHHCs (GFP-ZDHHCs) in HEK293T cells, with the FLAG-TRPV1 + GFP group serving as the negative control, β-actin serving as the loading control. One representative blot of three replicate experiments is shown. (C) Co-immunoprecipitation examined the interaction between GFP-ZDHHC4 and Flag-tagged N-terminal (Nt), transmembrane (TM), and C-terminal (Ct) domains of TRPV1 in HEK293T cells. Immunoprecipitation was performed using the anti-Flag antibody and analyzed by blotting with indicated antibodies. The full-length (FL) Flag-TRPV1 + ZDHHC4 group served as the positive control, and β-actin served as the loading control. One representative blot of three replicate experiments is shown. (D) Co-immunoprecipitation analysis validated the interaction between TRPV1 and ZDHHC4 in native DRG neurons, Co-immunoprecipitated samples was pulled down using anti-TRPV1 antibody (IgG as the negative control) and blotted with anti-TRPV1 and anti-ZDHHC4 antibodies, respectively. One representative blot of three replicate experiments is shown. (E) Immunoblotting displayed the protein levels of TRPV1, JAK2, phosphorylated JAK2 (p-JAK2), STAT3, phosphorylated STAT3 (p-STAT3), and ZDHHC4 in mouse DRG at various time points post intraplantar carrageenan injection. β-actin served as the loading control. One representative blot of three replicate experiments is shown. (F) The protein levels of TRPV1, p-JAK2, p-STAT3, and ZDHHC4 in (E) were quantified using gray intensity analysis ($n = 3$, $n$ represents biological replicates). (G) Representative confocal imaging (upper panels) and corresponding quantification analysis (lower panels) showing the expression and colocalization of ZDHHC4 (green) and TRPV1(red) in isolated DRG neurons from mice subjected to saline (left panels) or carrageenan (right panels) treatment. The lower panels illustrate the fluorescence intensity of ZDHHC4 and TRPV1, as measured along the yellow line that has been drawn across the cell. Fluorescence intensity was assessed by imageJ. Nuclei were stained with DAPI (blue). Scale bar, 10 µm. (H) Representative confocal imaging (upper panels) and corresponding statistical analysis (lower panels) showing the expression and colocalization of ZDHHC4 (green) and TRPV1(red) in DRGs from mice subjected to saline (left panels) or carrageenan (right panels) treatment; scale bar, 200 µm. The bars in the lower left panel illustrate the quantification of fluorescence intensity in the interaction area of the DRGs in mice that received either saline or carrageenan injections. The bars in the lower right panel quantify the extent of colocalization between TRPV1 and ZDHHC4 in the DRGs of saline- or carrageenan-injected mice ($n = 9$ cells for the saline group, $n = 9$ cells for the carrageenan group; $n$ represents biological replicates). The fluorescence intensity and proportion of interacting regions were analyzed using MicroscopX FINER. Each data point represents a separate biological replicate. The statistical significance was assessed using an unpaired Student's $t$-test. Data information: In (A, F, H), data were presented as mean ± SD. The $p$ values are displayed in the figure. n.s., not significant. Source data are available online for this figure.

JAK2 and STAT3 (p-JAK2 and p-STAT3), as well as ZDHHC4 protein levels, in DRGs of inflammatory pain mice induced by carrageenan. As expected, the boosted TRPV1 activity by inflammation indeed enhanced JAK2 and STAT3 phosphorylation (Fig. 2E,F). The p-JAK2 and p-STAT3 levels mirrored the changes of TRPV1 protein levels over the whole time-course of pain. On the other hand, ZDHHC4 protein levels were elevated upon STAT3 activation, and remained relatively constant thereafter (Fig. 2E,F). These observations were significantly reversed by inhibition of JAK2 activity by AG490, and were absent in $Trpv1^{-/-}$ mice treated with carrageenan (Appendix Fig. S2C,D). Moreover, through fluorescence imaging, we observed a concomitant increase of TRPV1 and ZDHHC4 protein levels, as well as their colocalization, in both DRG neurons and tissues from mice subjected to carrageenan treatment in comparison to mice treated with saline (Fig. 2G,H). Considering the role of ZDHHC4 in downregulating TRPV1 protein level, these findings likely reflected a feedback regulation mechanism to suppress the overactivation of TRPV1 and its subsequent pathways during pain relief. Overall, our data identified ZDHHC4 as a regulatory protein of TRPV1, being modulated by the JAK2-STAT3 signaling pathway.

## ZDHHC4 mediates TRPV1 palmitoylation and downregulates TRPV1 protein levels

Our data thus far reveal that TRPV1 degradation during pain resolution is accompanied by its palmitoylation, and the palmitoyl transferase ZDHHC4 physically interacts with TRPV1. We then sought to explore whether ZDHHC4 promoted TRPV1 palmitoylation and drove TRPV1 degradation. In ND7/23 cells, overexpression of ZDHHC4 remarkably enhanced the endogenous TRPV1 palmitoylation as demonstrated by the ABE assay (Fig. 3A). Furthermore, both the surface and total TRPV1 abundance were reduced by the overexpressed ZDHHC4 (Fig. 3B). These findings

reveal a negative correlation between the palmitoylation of TRPV1 and its protein levels, suggesting ZDHHC4 promotes TRPV1 palmitoylation for degradation.

We further reinforced our conclusion by silencing ZDHHC4 using small hairpin RNA (shRNA). The shRNA#1 exhibited robust knockdown of the endogenous ZDHHC4 as compared to the scramble control (shCon) and the shRNA#2 (Fig. 3C, upper panel). Moreover, shRNA#1 significantly reduced TRPV1 palmitoylation (Fig. 3C, lower panel). These observations strongly support ZDHHC4-dependent palmitoylation of TRPV1. The remaining TRPV1 palmitoylation after disrupting ZDHHC4 could be mediated by a compensatory effect of other ZDHHC isoforms or trace amounts of ZDHHC4 in cells. Given the role of ZDHHC4 in TRPV1 degradation, shRNA#1-mediated knockdown of ZDHHC4 led to an elevated TRPV1 protein level both on PM and in the cytosol (Fig. 3D). In HEK293T cells, coexpression of ZDHHC4 with TRPV1 also augmented TRPV1 palmitoylation and reduced TRPV1 protein levels (Fig. 3E,F), consistent with the inhibitory effect of ZDHHC4 on endogenous TRPV1 in ND7/23 cells. Collectively, these data suggest that ZDHHC4 negatively regulates TRPV1 protein levels by enhancing its palmitoylation.

## Coexpression of ZDHHC4 reduces TRPV1 activity

Having observed that ZDHHC4 downregulates TRPV1 protein levels, we proceeded to assess the functional consequence via electrophysiological assays. Considering the polymodal gating of TRPV1, we examined the effects of ZDHHC4 coexpression on the current density and biophysics of TRPV1 in response to capsaicin, acid, and heat stimuli. The sample traces in Fig. 4A illustrate the TRPV1 currents evoked by increasing concentrations of capsaicin, with or without ZDHHC4 coexpression. The resulting concentration-response curves demonstrate that neither the half-maximal effective concentrations ($EC_{50s}$) nor the Hill coefficients

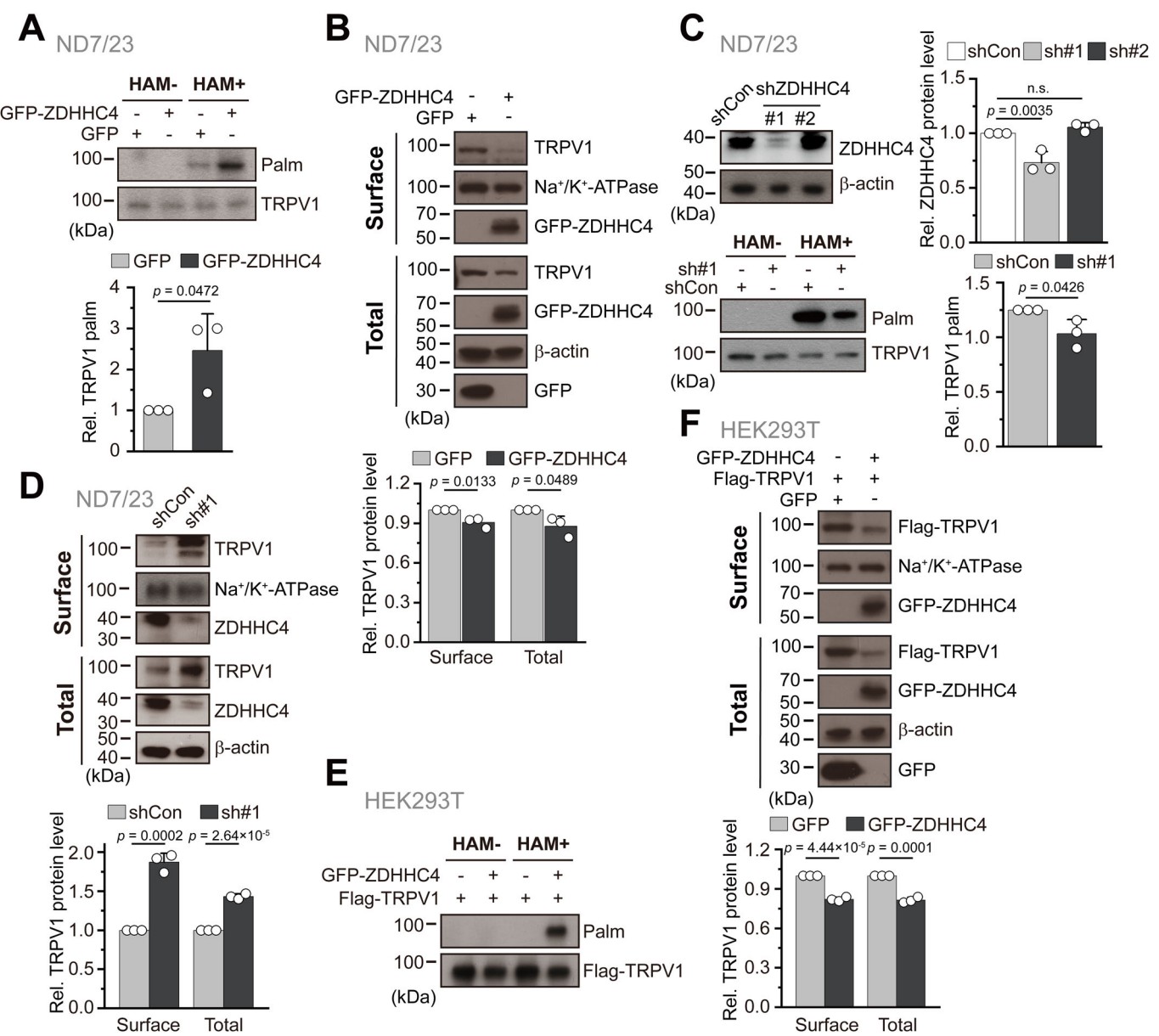

**Figure 3. ZDHHC4 mediates TRPV1 palmitoylation and downregulates TRPV1 protein levels.**

(A) ABE assay, followed by immunoblotting, was employed to analyze the palmitoylation of TRPV1 in ND7/23 cells with or without overexpression of GFP-ZDHHC4 (GFP as the transfection control), TRPV1 itself was used as the loading control. In the quantification bar graph, statistical significance was assessed using an unpaired Student's t-test with the resulting p value illustrated in the figure ($n = 3$, $n$ represents biological replicates). (B) Immunoblotting examining plasma membrane (upper panel) and total (lower panel) TRPV1 protein levels in ND7/23 cells with or without GFP-ZDHHC4 overexpression (upper panel). $Na^+/K^+$ ATPase and β-actin were used as the loading controls for plasma membrane and total proteins, respectively. In the quantification bar graph, statistical significance was assessed using an unpaired Student's t-test, and the resulting p values are shown in the figure ($n = 3$, $n$ represents biological replicates). (C) Upper left panel: ND7/23 cells transfected with ZDHHC4-targeting shRNA #1 (shZDHHC4 #1), but not #2 (shZDHHC4 #2), for 48 h exhibited remarkably reduced ZDHHC4 protein levels. The control shRNA (shCon) served as the transfection control, and β-actin was the loading control in immunoblotting ($n = 3$, $n$ represents biological replicates). Lower left panel: shZDHHC4 #1 significantly reduced TRPV1 palmitoylation in ND7/23 cells, as assessed by the ABE assay, compared to shCon. TRPV1 itself was the loading control ($n = 3$, $n$ represents biological replicates). In the quantification bar graphs in the right panel, the statistical significance was assessed using one-way ANOVA with post hoc Dunnett analysis (Rel. ZDHHC4 protein level) or Student's unpaired t-test (Rel. TRPV1 palm). (D) Immunoblotting demonstrated that shZDHHC4 #1 increased both the plasma membrane (upper panel) and total (lower panel) TRPV1 protein levels in ND7/23 cells, as compared with shCon RNA (upper panel). In the quantification bar graph, the statistical significance was assessed using an unpaired Student's t-test ($n = 3$, $n$ represents biological replicates). (E) and (F) replicated the analyses from panels (A) and (B), respectively, with the distinction that HEK293T cells lacking endogenous ZDHHC4 expression were used ($n = 3$, $n$ represents biological replicates). In the quantification bar graph in (F), statistical significance was assessed using an unpaired Student's t-test. Data information: In panels (A–D, F), each data point represents a separate biological replicate. In (A–D, F), data were presented as mean ± SD. The p values are illustrated in the figure. n.s., not significant. Source data are available online for this figure.

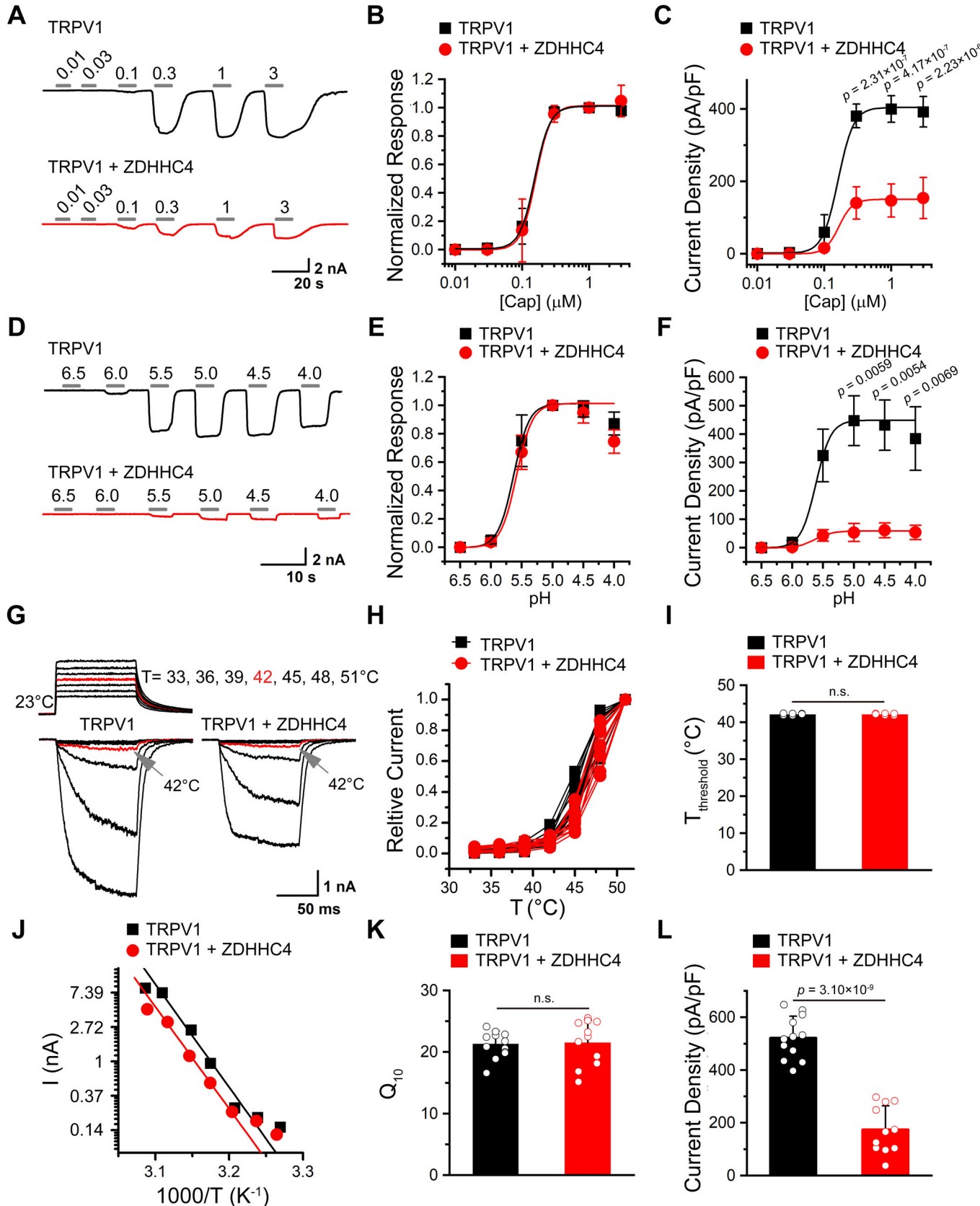

◄

**Figure 4.  ZDHHC4 reduces TRPV1 activity.**

(**A**) Family of representative TRPV1 currents evoked by capsaicin, under conditions with or without ZDHHC4 coexpression. (**B**) Normalized dose-response relationships of capsaicin, the $EC_{50}$s, and slope factors ($nHs$) for capsaicin activation were $0.15 \pm 0.004$ μM and $3.87 \pm 0.19$ for the TRPV1 group, and $0.16 \pm 0.006$ μM and $3.98 \pm 0.26$ for the TRPV1 + ZDHHC4 group. (**C**) Summary plot of current density in response to capsaicin. The statistical significance was assessed using an unpaired Student's $t$-test. (**D**) Family of representative TRPV1 currents evoked by acid. (**E, F**) normalized dose-response relationships of acid activation (**E**) and summary plot of current density in response to acid (**F**). The $pH_{50}$s and slope factors ($nHs$) for acid activation were $5.63 \pm 0.01$ and $3.42 \pm 0.17$ for the TRPV1 group and $5.59 \pm 0.01$ and $3.37 \pm 0.17$ for the TRPV1 + ZDHHC4 group, respectively. The statistical significance was assessed using an unpaired Student's $t$-test. (**G, H**) Family of representative currents activated by heat (**G**) and the resulting normalized current-temperature relationship (**H**). In (**H**), each curve represents measurements from an individual cell, normalized to its maximum responses at 51 °C, with temperature calibrated offline from the pipette current using the temperature dependence of electrolyte conductivity. (**I**) Summary bar graph showing that ZDHHC4 coexpression did not affect the threshold activation temperature ($T_{threshold}$) of TRPV1. $T_{threshold}$ was $42.14 \pm 0.06$ °C and $42.15 \pm 0.07$ °C for the TRPV1 and TRPV1 + ZDHHC4 groups, respectively. The statistical significance was assessed using an unpaired Student's $t$-test. (**J**) Arrhenius plot of steady-state currents for the two groups shown in (**G**). The major component of the reflection, representing the strong temperature dependence, was fitted to a linear equation. (**K**) The $Q_{10}$ values derived from the linear fits in the Arrhenius plot of steady-state currents were $21.31 \pm 0.64$ and $21.96 \pm 1.10$ for the TRPV1 and TRPV1 + ZDHHC4 groups, respectively. The statistical significance was assessed using an unpaired Student's $t$-test. (**L**) Summary current density bar graph showing that ZDHHC4 coexpression significantly reduced heat-evoked TRPV1 currents. The current was measured at 51 °C and current density was $522.35 \pm 23.58$ pA/pF and $174.31 \pm 27.08$ pA/pF for the TRPV1 and TRPV1 + ZDHHC4 groups, respectively. The statistical significance was assessed using an unpaired Student's $t$-test. Data information: In (**A–C**), $n = 7$ cells for TRPV1, $n = 6$ cells for TRPV1 + ZDHHC4; in (**D–F**), $n = 6$ cells for TRPV1, $n = 6$ cells for TRPV1 + ZDHHC4; in (**G–L**), $n = 12$ cells for TRPV1, $n = 11$ cells for TRPV1 + ZDHHC4; $n$ represents biological replicates. In panels (**I, K, L**), each data point represents a separate biological replicate. In (**B, C, E, F, I, K, L**), data were presented as mean ± SD. The $p$ values are illustrated in the figure. n.s., not significant. Source data are available online for this figure.

($n_H$) for capsaicin-activited TRPV1 were affected by ZDHHC4 (Fig. 4B). However, as anticipated, ZDHHC4 coexpression did remarkably reduce the current density of TRPV1 channels in response to capsaicin (Fig. 4C). Similar findings were observed for acid activation of TRPV1 (Fig. 4D–F). In the presence of ZDHHC4, acid of pH 5.5–4.0 activated much smaller whole-cell TRPV1 currents, while the current-pH relationships were not affected as demonstrated by unchanged $pH_{50}$ and proton sensitivity (reflected by the $n_H$ value). We next investigated whether ZDHHC4 affects the thermosensitivity of TRPV1 by utilizing an infrared laser diode heating system to precisely increase the local temperature of the recorded cell. As a result, rising temperature robustly evoked TRPV1 currents, with much smaller currents and decreased current density observed in the TRPV1 + ZDHHC4 group compared to the TRPV1-alone group (Fig. 4G,L). Moreover, neither the temperature threshold for activation ($T_{threshold}$) nor the temperature coefficient ($Q_{10}$) of TRPV1 were not affected by ZDHHC4 (Fig. 4H–K). Overall, these findings show that ZDHHC4 downregulates TRPV1 cellular response without affecting channel sensitivity to agonists or temperature, corroborating the reduction in surface TRPV1 protein levels as revealed by biochemical assays.

## ZDHHC4 regulation of TRPV1 requires its enzymatic activity

We next sought to explore whether ZDHHC4 downregulated TRPV1 protein levels by directly enhancing the channel's palmitoylation. We first tested if ZDHHC4 still maintains its regulatory effect on TRPV1 when its enzyme activity was deprived or pharmacologically inhibited. The ZDHHC4 truncation lacking the DHHC domain, referred to as ZDHHC4 (ΔDHHC), was stripped of its palmitoyl transferase activity (Zhao et al, 2022). Compared with the full-length (FL) ZDHHC4, the ZDHHC4 (ΔDHHC) truncation failed to palmitoylate TRPV1 (Fig. 5A). On the other hand, 2-bromopalmitate (2-BP), a nonmetabolizable palmitate analog that prevents the incorporation of palmitate into target proteins by palmitoyl transferase, also abrogated the palmitoylation of TRPV1 (Fig. 5B). Of note, TRPV1 protein levels remained relatively stable when its palmitoylation was diminished

(Fig. 5C,D). In alignment with these findings by biochemical assay, eliminating the enzyme activity of ZDHHC4 also overturned its inhibitory effect on TRPV1 current density, without affecting the channel sensitivity to capsaicin (Fig. 5E–G). Collectively, these data support the conclusion that the regulatory effect of ZDHHC4 on TRPV1 relies on its enzyme activity.

## TRPV1 is palmitoylated at C157, C362, C390, and C715 sites

Next we proceeded to identify the palmitoylation sites on TRPV1. S-palmitoylation occurs at the intracellular cysteine sites accessible to palmitoyl transferase. Sequence alignment of various orthologous TRPV1 proteins from different species (rat, mouse, human, zebrafish, and chicken) revealed seven conserved intracellular cysteines (C157, C257, C362, C386, C390, C715, and C741) (Fig. 6A). Indeed, substituting of C157, C362, C390, or C715 with alanine, but not the others, reduced the palmitoylation of TRPV1 by ZDHHC4 to different extents (Fig. 6B), and partially rescued TRPV1 downregulation (Appendix Fig. S3A). Among these, the C362A mutation exhibited the most profound effect (Fig. 6B; Appendix Fig. S3A). Moreover, measuring the functional protein levels of these mutants by their current density in response to saturated concentrations of capsaicin demonstrated that the C362A mutation resulted in the complete elimination of the regulatory effect of ZDHHC4 (Fig. 6C,D; Appendix Fig. S3B,C). Additionally, compared to the C257A, C386A, and C741A mutations, the C157A, C390A, and C715A mutations exhibited a partial rescue effect on reduction of TRPV1 activity by ZDHHC4 (Fig. 6C,D; Appendix Fig. S3B,C). None of the mutations significantly altered the channel's apparent affinity for capsaicin, regardless of whether ZDHHC4 expression was present or absent (Appendix Fig. S3D).

These findings motivated us to combine the effective mutations and make the quadruple cysteine mutant 4CA (C157A/362A/390A/715A). As a result, the 4CA TRPV1 mutant fully abandoned its regulation by ZDHHC4, as illustrated by the loss of ZDHHC4-mediated currents density reduction (Fig. 6E), protein levels downregulation (Fig. 6F,G), and protein palmitoylation (Fig. 6H). Thus, ZDHHC4 catalyzes TRPV1 palmitoylation at C157, C362,

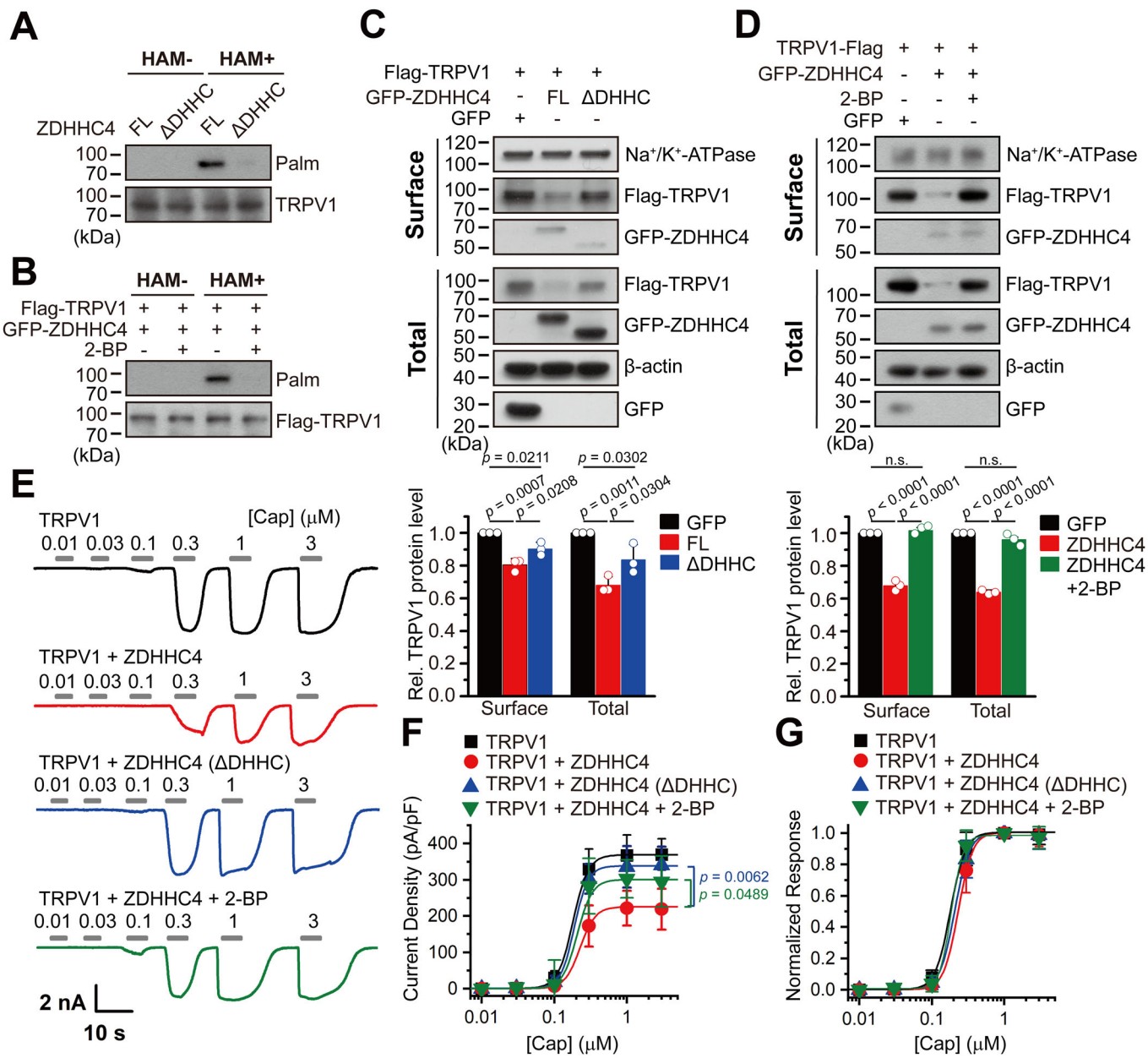

**Figure 5. ZDHHC4 enzyme activity is indispensable for TRPV1 regulation.**

(A) ABE assay followed by immunoblotting demonstrated that the ZDHHC4 truncation [ZDHHC4(ΔDHHC)] deprived of its enzyme activity lost the ability to catalyze TRPV1 palmitoylation in HEK293T cells as compared to the full-length (FL) ZDHHC4, TRPV1 itself was used as the loading control. (B) Treating the cells overnight with 100 μM 2-bromopalmitate (2-BP) to pharmacologically inhibit ZDHHC4 also eliminated the palmitoylation of TRPV1. (C) TRPV1 was co-transfected with ZDHHC4 or ZDHHC4(ΔDHHC) truncation, and immunoblotting was conducted to test the plasma membrane and total protein levels of TRPV1 (upper panels). GFP was used as the transfection control, and Na$^+$/K$^+$ ATPase and β-actin served as the loading control for surface and total proteins, respectively ($n = 3$, $n$ represents biological replicates). (D) TRPV1 was co-transfected with ZDHHC4 and treated with 2-BP overnight, GFP was used as the transfection control (upper panels) ($n = 3$, $n$ represents biological replicates). In (C, D), the lower panels illustrated the corresponding quantifications and statistical comparisons. Each data point represents a separate biological replicate. The statistical significance was assessed using one-way ANOVA with post hoc Turkey analysis. (E) Representative capsaicin (0.01–3 μM)-evoked TRPV1 currents in HEK293T cells under indicated conditions, showing that ZDHHC4(ΔDHHC) co-transfection did not affect the current amplitude as opposed to ZDHHC4, and 2-BP treatment rescued the reduction of the current by ZDHHC4. Currents were recorded at the holding potential of −60 mV. (F, G) Current density-concentration (F) and normalized response-concentration relationships (G) for capsaicin activation of TRPV1 in the four conditions shown in (E). Eliminating or inhibiting ZDHHC4's enzyme activity substantially impaired its ability to reduce TRPV1 current density without affecting the channel's capsaicin sensitivity. The EC$_{50}$ and Hill coefficient ($n_H$) values were: TRPV1 (EC$_{50}$ = 0.18 ± 0.002 μM, $n_H$ = 4.20 ± 0.09; $n = 7$ cells); TRPV1 + ZDHHC4 (EC$_{50}$ = 0.23 ± 0.03 μM, $n_H$ = 4.37 ± 0.20; $n = 6$ cells); TRPV1 + ZDHHC4(ΔDHHC) (EC$_{50}$ = 0.21 ± 0.003 μM, $n_H$ = 4.25 ± 0.12; $n = 6$ cells), and TRPV1 + ZDHHC4 + 2-BP (EC$_{50}$ = 0.11 ± 0.004 μM, $n_H$ = 2.32 ± 0.26; $n = 7$ cells). The statistical significance was assessed using one-way ANOVA with post hoc Dunnett analysis in (F). Data information: In (C, D, F, G), data were presented as mean ± SD. The $p$ values are illustrated in the figure. n.s., not significant. Source data are available online for this figure.

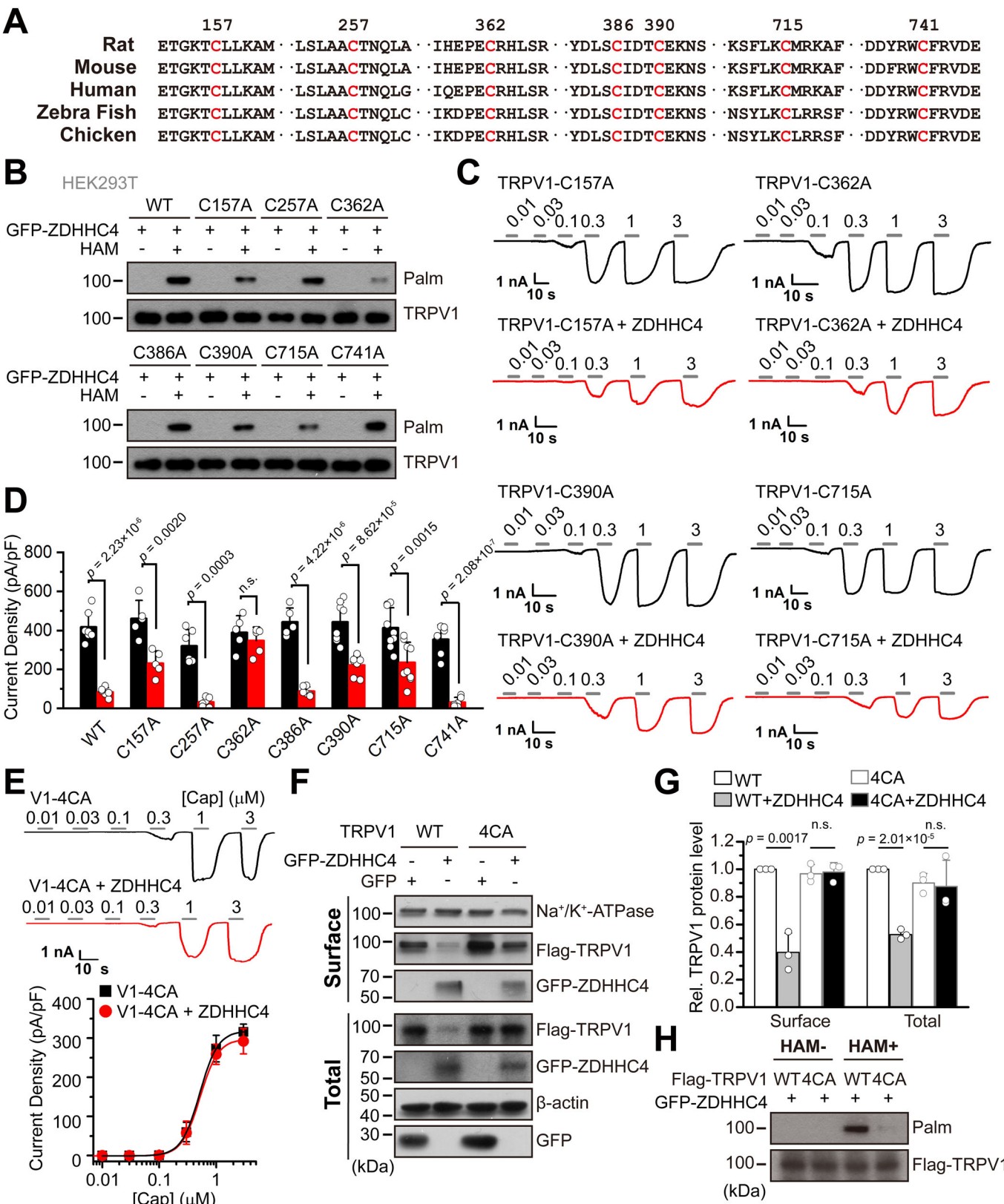

Figure 6. TRPV1 is palmitoylated at C157, C362, C390, and C715.

(A) Sequence alignment highlighting the conserved cysteine residues between orthologous TRPV1 channels from five different species. (B) ABE assay followed by immunoblotting illustrating the palmitoylation of wild-type (WT) TRPV1 and 7 different cysteine mutants as presented, showing that the C157A, C362A, C390A, and C715A mutants exhibited diminished palmitoylation compared with WT-TRPV1. (C) Representative capsaicin (0.01–3 μM)-evoked currents of the C157A, C362A, C390A, and C715A TRPV1 mutant channels in the absence (black) and presence (red) of ZDHHC4, currents were recorded at the holding potential of −60 mV. (D) Summary bar graph of the maximal capsaicin-evoked current density of WT-TRPV1 and seven cysteine mutants, in the absence and presence of ZDHHC4, emphasizing that the C362A mutation eliminated, and the C157A, C390A, and C715A mutations apparently reduced the regulation of TRPV1 by ZDHHC4. Each data point represents a separate biological replicate ($n = 5$–9 cells for each group). The statistical significance between groups was assessed using an unpaired Student's $t$-test. (E) Representative capsaicin (0.01–3 μM)-evoked currents (*upper panel*) and the resulting current density-concentration relationships (lower panel) of the TRPV1-4CA mutant harboring all the four cysteine mutations (C157A/362A/390 A/715A) in HEK293T cells in the absence and presence of ZDHHC4, showing ZDHHC4 did not affect the TRPV1-4CA currents. $EC_{50} = 0.50 \pm 0.01$ μM, $n_H = 2.56 \pm 0.08$ for TRPV1-4CA ($n = 9$ cells) and $EC_{50} = 0.48 \pm 0.007$ μM, $n_H = 2.80 \pm 0.08$ for TRPV1-4CA + ZDHHC4 ($n = 10$ cells). Currents recorded at −60 mV. (F, G) Immunoblotting (F) and the corresponding quantification via gray intensity analysis (G) depicting ZDHHC4 lost its regulatory effects on both the plasma membrane and total protein levels of TRPV1-4CA mutant channel in HEK293T cells, as opposed to WT-TRPV1. GFP was the transfection control, and Na$^+$/K$^+$ ATPase and β-actin were the loading control of surface and total proteins, respectively ($n = 3$, $n$ represents biological replicates). In (G), the statistical significance was assessed using an unpaired $t$-test. (H) The TRPV1-4CA mutant channel was insensitive to palmitoylation catalyzed by ZDHHC4 in HEK293T cells. Data information: In (D, G), each data point represents a separate biological replicate. In (D, E, G), data were presented as mean ± SD. The $p$ values are illustrated in the figure. n.s., not significant. Source data are available online for this figure.

C390, and C715 sites. These data provide further evidence supporting that ZDHHC4 palmitoylates TRPV1 for its degradation.

## TRPV1 palmitoylation is counterbalanced by the depalmitoylase APT1

Protein palmitoylation is a reversible post-translational modification governed by palmitoyl transferases and depalmitoylases. After uncovering the aforementioned ZDHHC4 regulation of TRPV1, we sought to pinpoint the depalmitoylase that counteracts this process.

Typically, depalmitoylase can be categorized into three types: acyl-protein thioesterases (APT1, APT2), palmitoyl-protein thioesterases (PPT1, PPT2), and alpha/beta hydrolase domain-containing protein 17A/B/C (ABHD17A/B/C) (Won et al, 2017). Among them, APTs and PPTs exhibit a stronger association with neural tissues (Mondal et al, 2022; Shen et al, 2022; Yuan et al, 2021). We thus examined the protein levels of APTs and PPTs in the neuron-derived ND7/23 cells. Immunoblotting analysis confirmed the presence of APT1 and APT2, but not PPTs, in ND7/23 cells, while APT1 showed much higher abundance (Fig. 7A). This suggested the possible involvement of APT1 and APT2 in mediating TRPV1 depalmitoylation. Further, we conducted knockdown of APT1 and APT2, respectively, using their shRNAs and examined the effects on TRPV1 protein levels. Intriguingly, we observed a decrease in TRPV1 protein levels by APT1 but not APT2 knockdown (Fig. 7B,C). Moreover, TRPV1 palmitoylation was also enhanced by reducing APT1 protein level (Fig. 7D). Notably, coexpression of APT1 together with ZDHHC4 almost fully over-whelmed its downregulation of TRPV1 functional protein levels as reflected by the capsaicin-evoked TRPV1 currents (Fig. 7E,F). As for ZDHHC4, APT1 coexpression did not affect TRPV1 apparent affinity for capsaicin (Fig. 7G). These findings support that APT1 depalmitoylates TRPV1, thereby counterbalancing ZDHHC4-mediated TRPV1 palmitoylation. The functionally balanced signaling cascades thus help to restore TRPV1 protein levels to the equilibrium states.

## Manipulating TRPV1 palmitoylation affects pain resolution in vivo

To assess the impact of TRPV1 palmitoylation on the resolution of inflammatory pain, we used adeno-associated viruses (AAV) to express shRNAs, targeting ZDHHC4 and APT1, in DRG neurons

of wild-type and $Trpv1^{-/-}$ mice. The control AAV (AAV-Scramble), AAV encoding ZDHHC4 knockdown shRNA (AAV-shZDHHC4), or AAV encoding APT1 knockdown shRNA (AAV-shAPT1) was administered intrathecally at the L5–L6 lumbar region, respectively. These shRNAs were successfully delivered into both the DRGs and the spinal cord, as evidenced by the abundant expression of marker genes encoding the enhanced green fluorescent protein (Fig. 8A; Appendix Fig. S4A). In comparison to AAV-shScramble, treatments with AAV-shZDHHC4 and AAV-shAPT1 demonstrated effective downregulation of ZDHHC4 and APT1 protein levels, respectively, in DRGs from both wild-type and $Trpv1^{-/-}$ mice (Fig. 8B). In line with our in vitro experiments using ND7/23 cells, the palmitoylation of TRPV1 was found to be decreased by AAV-shZDHHC4 and increased by AAV-shAPT1 (Appendix Fig. S4B), while TRPV1 protein levels were observed to be upregulated by AAV-shZDHHC4 and downregulated by AAV-shAPT1 (Fig. 8B). These biochemical findings are consistent with the observation that AAV-shZDHHC4 significantly increased TRPV1 current density in DRG neurons, without affecting the channel's sensitivity to capsaicin (Fig. 8C).

Two weeks post AAV injection, mice were then injected with carrageenan at the plantar region to induce inflammatory pain. The pain behavior and TRPV1 protein levels in DRGs were monitored for a period of 0-14 days following the carrageenan injection. As a result, ZDHHC4 knockdown led to slower TRPV1 degradation over time in comparison to the scramble control. However, this was significantly reversed by APT1 knockdown (Appendix Fig. S4C,D). Importantly, the knockdown of ZDHHC4 or APT1 in DRG slowed or accelerated the recovery from inflammatory pain in wild-type mice (Fig. 8D, left panels) but not in $Trpv1^{-/-}$ mice (Fig. 8D, right panels) in the mechanical allodynia tests. Neither wild-type nor $Trpv1^{-/-}$ mice exhibited alterations in the baseline mechanical pain threshold following saline treatment (Appendix Fig. S4E). In wild-type mice treated with carrageenan, thermal hyperalgesia exhibited a more rapid resolution than mechanical allodynia (Figs. 8E, upper left panel and 1B). Nevertheless, the knockdown of ZDHHC4 or APT1 exhibited a slight yet statistically significant impact on the resolution of thermal hyperalgesia in wild-type mice with inflammatory pain (Fig. 8E, left panel). Baseline thermal hyperalgesia was also reduced or increased by ZDHHC4 or APT1 knockdown in wild-type mice subjected to saline treatment (Appendix Fig. S4F, left panels). These effects on thermal

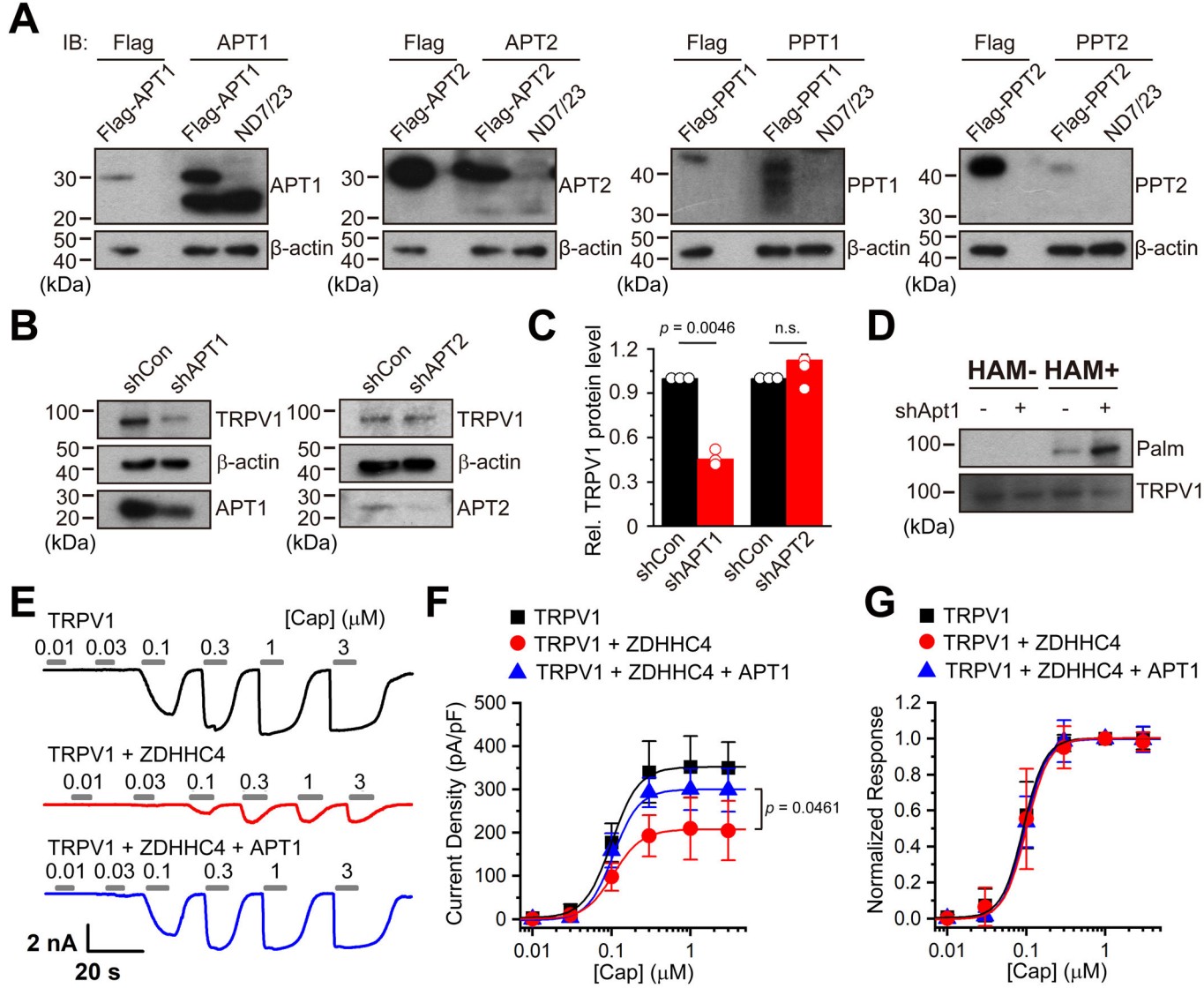

**Figure 7. APT1 mediates the depalmitoylation of TRPV1.**

(A) Immunoblotting assessing the protein levels of APT1, APT2, PPT1, and PPT2 in ND7/23 cells, FLAG-tagged APT1, APT2, PPT1, and PPT2 expressed in HEK293T served as the positive control and were detected using their specific antibodies as well as the anti-FLAG antibody. (B, C) Representative blotting (B) and related statistical analysis (C) demonstrated knocking down APT1, but not APT2 with their respective shRNAs (shAPT1 and shAPT2) significantly reduced TRPV1 protein levels in ND7/23 cells. The analysis was performed 48 h after shRNA transfection, and cells transfected with control shRNA (shCon) served as the negative control ($n = 3$, $n$ represents biological replicates). Each data point represents a separate biological replicate. The statistical significance was assessed using an unpaired Student's $t$-test. (D) ABE assay followed by immunoblotting confirmed knocking down APT1 with shAPT1 enhanced TRPV1 palmitoylation in ND7/23 cells. (E) Representative capsaicin (0.01–3 μM)-evoked currents in HEK293T cells transfected with TRPV1 alone, TRPV1 + ZDHHC4, or TRPV1 + ZDHHC4 + APT1. (F, G) Current density-concentration (F) and normalized response-concentration (G) relationships of capsaicin activating TRPV1 under conditions shown in (E). APT1 coexpression reversed the ZDHHC4-induced reduction in TRPV1 current density without affecting the channel's capsaicin sensitivity, the $EC_{50}$ and Hill coefficient ($n_H$) values were: TRPV1 ($EC_{50} = 0.09 \pm 0.01$ μM, $n_H = 3.16 \pm 0.23$; $n = 6$ cells); TRPV1 + ZDHHC4 ($EC_{50} = 0.10 \pm 0.001$ μM, $n_H = 3.05 \pm 0.23$; $n = 6$ cells); TRPV1 + ZDHHC4 + APT1 ($EC_{50} = 0.10 \pm 0.001$ μM, $n_H = 3.21 \pm 0.13$; $n = 6$ cells). The statistical significance was assessed using an unpaired Student's $t$-test in (F). Data information: In (C, F, G), data were presented as mean ± SD. The $p$ values are illustrated in the figure. n.s., not significant. Source data are available online for this figure.

hyperalgesia were not observed in *Trpv1⁻/⁻* mice subjected to either carrageenan or saline treatment (Fig. 8E, right panels; Appendix Fig. S4F, right panels).

Moreover, under both normal and inflammatory physiological conditions, ZDHHC4 knockdown mice exhibited enhanced nocifensive behavior in response to intraplantar capsaicin injection, evidenced by increased time spent lifting, licking, and biting the

injected foot. Conversely, APT1 knockdown mice showed markedly attenuated nocifensive behaviors (Fig. 8F,G). This reflects that the knockdown of ZDHHC4 or APT1 disrupts the balance between TRPV1 palmitoylation and depalmitoylation, influencing TRPV1 protein levels and ultimately altering mouse sensitivity to capsaicin. Therefore, manipulating TRPV1 palmitoylation in vivo modulates inflammatory pain recovery.

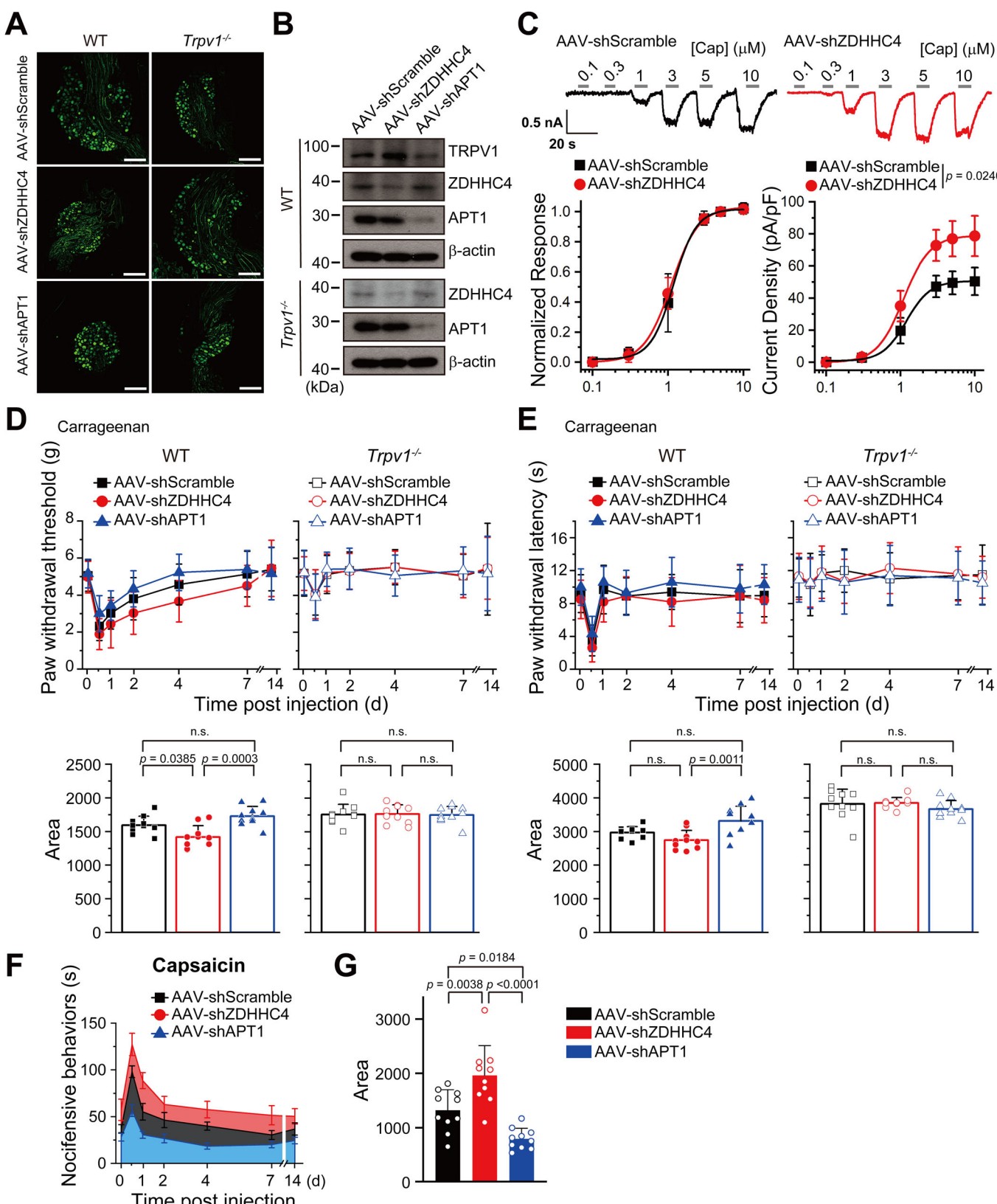

**Figure 8.   ZDHHC4 and APT1 regulate inflammatory pain resolution by modulating TRPV1 in vivo.**

(A) GFP fluorescence confirmed successful transduction of the indicated AAVs in DRGs of both wild-type (WT) and $Trpv1^{-/-}$ mice. Scale bar: 200 μm. (B) Immunoblotting demonstrating that AAV-shZDHHC4 or AAV-shAPT1 mediate successful knockdown of ZDHHC4 and APT1 in DRGs of both WT (*left panel*) and $Trpv1^{-/-}$ (right panel) mice, leading to upregulation or downregulation of TRPV1 in WT mice, respectively. AAV-shScramble served as the control, and β-actin was used as the loading control in immunoblotting. (C) Representative TRPV1 currents evoked by varying concentrations of capsaicin in DRG neurons from mice treated with either AAV-shScramble ($n = 6$ cells) or AAV-shZDHHC4 ($n = 6$ cells) (upper panel). The normalized response-concentration (lower left panel) and current density-concentration (lower right panel) relationships showed that ZDHHC4 knockdown significantly increased TRPV1 current density without affecting the channel's capsaicin sensitivity. Statistical significance was assessed using an unpaired Student's *t*-test. (D, E) Paw withdrawal threshold and latency of WT and $Trpv1^{-/-}$ mice infected with indicated AAVs. Inflammatory pain was induced via intraplantar injection of carrageenan two weeks post AAV infection, with the $Trpv1^{-/-}$ group serving as the control ($n = 10$ mice for each group). Pain behavior was monitored over a 14-day period. Upper panels showed the time-dependent change in paw withdrawal threshold (D) and latency (E); Lower panels illustrated the corresponding AUC (area under curve) calculated from the curve shown in the upper panels. Statistical significance was assessed using one-way ANOVA with post hoc Turkey analysis. (F, G) The time-dependent development of capsaicin-evoked (2 μg, through intraplantar injection) nocifensive behaviors in WT mice infected with the indicated AAVs was observed over a 14-day period following carrageenan administration. Panel (F) shows this development and the calculated AUC provides a quantitative assessment (G); compared with the AAV-Scramble control, AAV-shZDHHC4 markedly increased, whereas AAV-shAPT1 remarkably decreased nocifensive behaviors ($n = 10$ mice for each group). AAVs were administered via intrathecal injection in these experiments. Statistical significance was assessed using one-way ANOVA with post hoc Turkey analysis. Data information: In summary bar graphs in panel (D, E, G), each data point represents a separate biological replicate. In (C–G), data were presented as mean ± SD. The *p* values are illustrated in the figure. n.s., not significant. Source data are available online for this figure.

# Discussion

In the present study, we unveil that the palmitoyl acyltransferase ZDHHC4 mediates the palmitoylation of TRPV1 and its degradation, which contributes to the resolution of inflammatory pain. Appendix Fig. S5 summarizes the dynamic regulation of TRPV1 during pain relief: enhanced TRPV1 activity correlates with increased JAK2-STAT3 signaling, triggering phosphorylated STAT3 (p-STAT3) translocation to nuclei to initiate ZDHHC4 expression; ZDHHC4 physically interacts with and palmitoylates TRPV1, promoting its degradation for pain relief; APT1 counterbalances TRPV1 palmitoylation, stabilizing it on the plasma membrane so as to reinstate the body sensory states.

The pivotal role of TRPV1 in pain perception has been well established during the past decades and the regulatory mechanisms of TRPV1 in pain development have been delineated (Iftinca et al, 2021). Pharmacologically inhibiting TRPV1 with its antagonists is deemed a promising strategy for pain management, however, it causes several unwanted side effects like hyperthermia that constrains its clinical application (Iftinca et al, 2021). One alternative strategy for pain treatment is to intervene in the molecular pathways regulating TRPV1 function during pathological processes. However, in contrast to the wide investigation of TRPV1 hyperfunction in pain development, little is known as to how the function of TRPV1 is attenuated during pain relief.

Inflammations boost TRPV1 function by upregulating its protein levels and sensitizing it to stimuli via the action of inflammatory cytokines, ROS, and acid. In the present study, we uncover that the palmitoyl transferase ZDHHC4 mediates TRPV1 palmitoylation and its downregulation, which promotes inflammatory pain relief. Protein palmitoylation is a widespread lipid modification in which one or more cysteines on a substrate protein are covalently modified by palmitoyl group via forming a thioester, regulating for instance protein stability, activity, trafficking, subcellular localization, and protein-protein interaction (Chamberlain and Shipston, 2015). A growing number of human diseases are associated with the dysregulation of palmitoylation, including neurological disorders and several types of cancer (Chavda et al, 2014; Fukata and Fukata, 2010; Ko and Dixon, 2018). Our study uncovers that TRPV1 is subjected to palmitoylation during the pain relief phase, which plays a role in controlling the time window of animal nociceptive sensation. These data thus update our knowledge on the

post-translational modification of the TRPV1 channel. Moreover, in contrast to phosphorylation and SUMOylation of TRPV1, which promotes hyperalgesia by sensitizing the channel to various gating stimuli (Planells-Cases et al, 2005; Wang et al, 2018), palmitoylation modification downregulates TRPV1 protein levels both in the cytosol and on PM. We have observed that the upregulated ZDHHC4 protein levels persisted for several days after the induction of inflammatory pain (Fig. 2E,F), likely reflecting a post hoc adaptation required for inflammatory pain relief. Additionally, we have observed that the palmitoylated TRPV1 is degraded through the lysosomal pathway. This resembles the effects of intense stimulations, which lead to the TRPV1 trap within the endosomal and lysosomal networks under the regulation of synaptotagmin 7 (Sanz-Salvador et al, 2012; Tian et al, 2019). Both mechanisms aim to downregulate TRPV1 protein levels to prevent excessive pain. It is worth investigating whether there is a crosstalk between these two regulatory pathways.

The partial effect of ZDHHC4 knockdown on pain relief in vivo suggests other parallel mechanisms being implicated in this process. Recently, interferon-stimulated genes by neuronal STING immune regulators have also been shown to pain resolution (Defaye et al, 2024). In this model, the JAK-STAT pathway is recruited in association with TRPV1 downregulation, as noted in the current study. Our finding that ZDHHC4-mediated palmitoylation contributes to TRPV1 degradation, provides further a mechanistic basis underlying TRPV1 functional regulation during inflammatory pain resolution. In our in vivo experiments, wild-type but not $Trpv1^{-/-}$ mice treated with carrageenan developed mechanical allodynia (Fig. 1A$_1$), implying that TRPV1 is involved in the process of mechanical nociception. Nonetheless, the role of the TRPV1 channel in mechanical allodynia has been in deliberation (Caterina et al, 2000; Davis et al, 2000; Mo et al, 2022; Stemkowski et al, 2016; Yin et al, 2013), raising the need for further studies to evaluate the specific contribution of TRPV1 to mechanical hypersensitivity.

Palmitoylation has been shown to enhance the hydrophobicity of proteins, facilitating their anchoring on PM (Shahinian and Silvius, 1995). In previous studies, protein palmitoylation mediated by ZDHHC enzymes has been found to be involved in the regulation of pain perception. For example, AKAP150 and its palmitoylation mediated by ZDHHC2 play a role in AMPA receptor-dependent modulation of nociceptive transmission (Li et al, 2021). Following palmitoylation by ZDHHC3, δ-catenin forms a complex with KIF3A and Nav1.6, thereby enhancing the

transmission of mechanical and nociceptive signals (Zhang et al, 2018). Except for pain regulation, palmitoylation is also involved in the membrane localization, subcellular localization, and voltage or ligand sensitivity of ion channels. TRPM7 undergoes palmitoylation at a cysteine cluster in its TRP domain, governing its ER exit and cellular distribution; ZDHHC17 in the Golgi and ZDHHC5 at the membrane are responsible for this modification, impacting TRPM7-mediated calcium uptake (Gao et al, 2022). Voltage-gated sodium channel (VGSC) β1 subunit is palmitoylated at Cys 162, which sustains the protein level at the plasma membrane, thus modulating sodium current (Bouza et al, 2020). In our study, ZDHHC4-mediated palmitoylation of TRPV1 leads to the channel degradation. These results are consistent with studies demonstrating palmitoylation could also mediate the lysosomal sorting and degradation of proteins (Bagh et al, 2017; Du et al, 2021). The detailed mechanisms guiding palmitoy-labeled TRPV1 proteins to lysosomal degradation await further exploration. Our electrophysiological experiments also show that the palmitoylation of TRPV1 does not affect its sensitivity to heat, acid, and capsaicin. By conducting a screening analysis of all cysteine residues on TRPV1, we also have demonstrated that palmitoylation at the C157, C362, C390, and C715 sites exclusively regulates TRPV1 activity without affecting the channel gating. This is different from what was observed in other ion channels, in which palmitoylation at different cysteine sites exhibits diverse effects on channel function. For example, S-palmitoylation at sites Cys1169 and Cys1170 on Nav1.6 modulates the channel's voltage dependence of inactivation, whereas the Cys1978 palmitoylation enhances Nav1.6 current (Pan et al, 2020).

Controlling the pain relief phase helps to confine the duration of body nociceptive suffering. Our data shed light on the molecular mechanisms regulating TRPV1 activity during pain resolution, suggesting that targeting TRPV1 palmitoylation could mitigate clinically intractable chronic pain, such as neuropathic pain, cancer-related pain, and various forms of chronic inflammatory pain.

# Methods

### Reagents and tools table

| Reagent/resource | Reference or source | Identifier or catalog number |
|---|---|---|
| **Experimental models** | | |
| HEK293T cells | ATCC | Cat# CRL-3216 |
| ND7/23 cells | The European Collection of Authenticated Cell Cultures (ECACC) | Cat# 92090903 |
| C57BL/6J mice | Hubei Province Center for Disease Control and Prevention | N/A |
| Trpv1$^{-/-}$ mice | GemPharmatech | C57BL/6JGpt-Trpv1em1Cd116d1025/Gpt |
| **Recombinant DNA** | | |
| TRPV1 cDNA (Rattus norvegicus) | David Julius et al | N/A |
| mouse ZDHHC2 cDNA | MiaoLingBio | Cat# p22652 |
| mouse ZDHHC3 cDNA | MiaoLingBio | Cat# p21495 |
| mouse ZDHHC4 cDNA | MiaoLingBio | Cat# P12421 |

| Reagent/resource | Reference or source | Identifier or catalog number |
|---|---|---|
| mouse ZDHHC12 cDNA | MiaoLingBio | Cat# p21829 |
| mouse ZDHHC15 cDNA | MiaoLingBio | Cat# p21490 |
| mouse ZDHHC17 cDNA | MiaoLingBio | Cat# p48999 |
| mouse ZDHHC18 cDNA | MiaoLingBio | Cat# p23780 |
| rat ZDHHC4 cDNA | Genscript | Cat# ORa03457 |
| rat APT1 | MiaoLingBio | Cat# P40923 |
| rat APT2 | MiaoLingBio | Cat# P30534 |
| rat PPT1 | MiaoLingBio | Cat# P32149 |
| rat PPT2 | MiaoLingBio | Cat# P12421 |
| pNsfGFP vector | Yu Ding, Fudan University, China | N/A |
| pCsfGFP vector | Yu Ding, Fudan University, China | N/A |
| **Antibodies** | | |
| Rabbit anti-TRPV1 | Alomone Labs | Cat#ACC-030 |
| Rabbit anti-ZDHHC4 | Abclonal | Cat#A17980 |
| Rabbit anti-JAK2 | Cell Signaling Technology | Cat#3230 |
| Rabbit anti-phospho-JAK2 | Abcam | Cat#AB32101 |
| Rabbit anti-STAT3 | Cell Signaling Technology | Cat#4904 |
| Rabbit anti-phospho-STAT3 | Cell Signaling Technology | Cat#9131 |
| Rabbit anti-APT1 | Proteintech | Cat#16055-1AP |
| Rabbit anti-APT2 | ABclonal | Cat# A15792 |
| Rabbit anti-PPT1 | Proteintech | Cat# 29653-1AP |
| Rabbit anti-PPT2 | Proteintech | Cat# 15429-1AP |
| HRP-streptavidin | Merck | Cat#RABHRP3 |
| Rabbit anti-Na$^+$/K$^+$ ATPase | Abcam | Cat#ab76020 |
| Rabbit anti-Flag | Proteintech | Cat#20543-1-AP |
| Rabbit anti-GFP | Solarbio Life Science | Cat#B1025F |
| Goat anti-mouse IgG (H + L) | Jackson Immunoresearch | Cat#115-035-003 |
| Goat anti-rabbit IgG (H + L) | Jackson Immunoresearch | Cat#111-005-003 |
| **Oligonucleotides and other sequence-based reagents** | | |
| qRT-PCR primers for DHHC1-F GCGCATGTCATTGAAGACCTGC | TSINGKE | N/A |
| qRT-PCR primers for DHHC1-R CACGCAGTTGTTGAGCCACTTG | TSINGKE | N/A |
| qRT-PCR primers for DHHC2-F AGTTCTGAGGCGAGCAGCCAAA | TSINGKE | N/A |
| qRT-PCR primers for DHHC2-R CAGACGGAACAATGATGACAGCG | TSINGKE | N/A |
| qRT-PCR primers for DHHC3-F TGGTGGGATTCCACTTCCTGCA | TSINGKE | N/A |
| qRT-PCR primers for DHHC3-R GCCTCAAAGCACAGCAGGATGA | TSINGKE | N/A |
| qRT-PCR primers for DHHC4-F AGGTGTGCTCCAGGGTAATC | TSINGKE | N/A |
| qRT-PCR primers for DHHC4-R CAGGTGCAAGACGATGAAGG | TSINGKE | N/A |
| qRT-PCR primers for DHHC5-F ACACACCTCAGCCTGGCTACTA | TSINGKE | N/A |
| qRT-PCR primers for DHHC5-R ATGGCGGCTGATGTGCTACTGC | TSINGKE | N/A |
| qRT-PCR primers for DHHC6-F AGTCTGCCAAGCATACAAGGCG | TSINGKE | N/A |
| qRT-PCR primers for DHHC6-R CAACAGGAGGAAGAGCGTGAAC | TSINGKE | N/A |

| Reagent/resource | Reference or source | Identifier or catalog number |
|---|---|---|
| qRT-PCR primers for DHHC7-F GCTCTGTCTTCGGTTCATGCTC | TSINGKE | N/A |
| qRT-PCR primers for DHHC7-R CTCAAGGCACAGGAAGACCAAC | TSINGKE | N/A |
| qRT-PCR primers for DHHC8-F GTGCCCTATCAGTACAGAGGAC | TSINGKE | N/A |
| qRT-PCR primers for DHHC8-R GGTGCTGTCATCTGCCAGAGTA | TSINGKE | N/A |
| qRT-PCR primers for DHHC9-F CTGCTGTGAAGTGCTTTGTGGC | TSINGKE | N/A |
| qRT-PCR primers for DHHC9-R TCTGTGGCAACAGGCTACTGCT | TSINGKE | N/A |
| qRT-PCR primers for DHHC11-F CATCCAGCAGAGGAGAAAGAGC | TSINGKE | N/A |
| qRT-PCR primers for DHHC11-R TCGGCGAAAGAGTAGACACTGG | TSINGKE | N/A |
| qRT-PCR primers for DHHC12-F TCACTCTGGTGCTCTTCCTGCA | TSINGKE | N/A |
| qRT-PCR primers for DHHC12-R GCTTAGCACCAGGAGCAGGAAT | TSINGKE | N/A |
| qRT-PCR primers for DHHC13-F TGGTTCTAGCCTGGACATCCGA | TSINGKE | N/A |
| qRT-PCR primers for DHHC13-R CCATCGCCAAAGCCGAAACTGT | TSINGKE | N/A |
| qRT-PCR primers for DHHC14-F ACAGAAGAGGCTATGTCCAGCC | TSINGKE | N/A |
| qRT-PCR primers for DHHC14-R GCTCTGAATGCACTGGTCTTGG | TSINGKE | N/A |
| qRT-PCR primers for DHHC15-F AGAGACCTGAGGTCCAGAAGCA | TSINGKE | N/A |
| qRT-PCR primers for DHHC15-R AGACAGAACAGTGATGGCAGCG | TSINGKE | N/A |
| qRT-PCR primers for DHHC16-F TGATGCTGCCTTTGAGCCTGTC | TSINGKE | N/A |
| qRT-PCR primers for DHHC16-R ACCGAGTAGGTTCGGAGGATGA | TSINGKE | N/A |
| qRT-PCR primers for DHHC17-F CTTCCTTGCCAACAGCGTTGCT | TSINGKE | N/A |
| qRT-PCR primers for DHHC17-R TGAGGTCCAGACTTCCAGTCTC | TSINGKE | N/A |
| qRT-PCR primers for DHHC18-F GGAGACGGAACTACCGCTTCTT | TSINGKE | N/A |
| qRT-PCR primers for DHHC18-R GCTGGTGTCTTTTTCAGAGCGG | TSINGKE | N/A |
| qRT-PCR primers for DHHC19-F CGAGCGTGTTTGCTGCCTTCAA | TSINGKE | N/A |
| qRT-PCR primers for DHHC19-R AGGTGAGGATGAAGAGTGGTCC | TSINGKE | N/A |
| qRT-PCR primers for DHHC20-F GCAAACCAGAGTGACTACGTCAG | TSINGKE | N/A |
| qRT-PCR primers for DHHC20-R CAGCTCCATTCTCTAGCCACTG | TSINGKE | N/A |
| qRT-PCR primers for DHHC21-F CTGAGCTGCTTACTTGCTACGC | TSINGKE | N/A |
| qRT-PCR primers for DHHC21-R TGCCCATGAAGGCAGCTAGTCT | TSINGKE | N/A |
| qRT-PCR primers for DHHC22-F CTTACGCTCCTGCCCACTTCAA | TSINGKE | N/A |
| qRT-PCR primers for DHHC22-R CGGAGGATCAACAGCAGTTGGT | TSINGKE | N/A |
| qRT-PCR primers for DHHC23-F GGATATGCGGTATCTGTGTACGG | TSINGKE | N/A |
| qRT-PCR primers for DHHC23-R GGTCAGCGATATTCCGTAAACCG | TSINGKE | N/A |
| qRT-PCR primers for DHHC24-F TCTACACAGTGGCTCTCCTGCT | TSINGKE | N/A |

| Reagent/resource | Reference or source | Identifier or catalog number |
|---|---|---|
| qRT-PCR primers for DHHC24-R AAAAGCAGCCCAGCACCACACA | TSINGKE | N/A |
| shRNA#1 for ZDHHC4 GGTGCTCCACCTGTGACTTAA | TSINGKE | N/A |
| shRNA#2 for ZDHHC4 GCCTAGTGGCAGTGTCAAATC | TSINGKE | N/A |
| Scramble shRNA for ZDHHC4 GCAACCTTCAAGAGGTCTTTA | TSINGKE | N/A |
| shRNA for APT1 GATGTACACACAGCACGATGG | TSINGKE | N/A |
| Scramble shRNA for APT1 GGAAGGGACCACCTTCAAAT | TSINGKE | N/A |
| shRNA for APT2 AAGAAATTCCTTCACAGCTGC | TSINGKE | N/A |
| Scramble shRNA for APT2 GGGCTCTTTGACAGTTCATAC | TSINGKE | N/A |
| **Chemicals, Enzymes and other reagents** | | |
| Anti-Flag Affinity Gel | Bimake | Cat#B23100 |
| HPDP-biotin | APE×BIO | Cat#A8008 |
| EZ-Link Sulfo-NHS-LC-Biotin | Thermo Fisher Scientific | Cat#21335 |
| High Capacity NeutrAvidin™ Agarose beads | Thermo Fisher Scientific | Cat#29202 |
| Puromycin | Merck | Cat#P8833 |
| 2-BP | Merck | Cat#238422 |
| Paraformaldehyde | Biosharp | Cat#BL539A |
| DAPI | Merck | Cat#D5492 |
| TRIzol | Takara | Cat# T9108 |
| DMEM | Gibco | Cat# 11965092 |
| FBS | Gibco | Cat# A5670701 |
| Lipofectamine 2000 | Invitrogen | Cat# 11668500 |
| AAV virus | Brain Case | N/A |
| **Software** | | |
| QStudio | Molecular Devices | N/A |
| ImageJ | National Institutes of Health | https://imagej.nih.gov/ij/ |
| IGOR | Wavemetrics | https://www.wavemetrics.com/ |
| SigmaPlot | SPSS Science | https://sigmaplot.en.softonic.com/ |
| OriginPro | OriginLab Corporation | https://www.originlab.com/ |
| **Other** | | |
| All-in-One cDNA Synthesis SuperMix | Biotool | Cat# B24101 |
| 2 × SYBR Green Fast qRT-PCR Master Mix | Biotool | Cat# B21704 |
| Axopatch 200B amplifier | Molecular Devices | N/A |
| BNC-2090/MIO acquisition system | National Instruments | N/A |
| borosilicate glass capillaries | World Precision Instruments | Cat# 1B150F-4 |
| Plantar Test Instrument | Ugo Basile | Cat# 19316 |
| Von Frey apparatus | Ugo Basile | Cat# 19316 |

## qRT-PCR

To evaluate the expression of different ZDHHCs in DRG neurons by qRT-PCR, total RNA was extracted from DRG neurons using TRIzol (Takara, T9108), and the first-strand cDNA was reverse-

transcribed with All-in-One cDNA Synthesis SuperMix (Biotool). ZDHHCs' expression was examined in a Bio-Rad CFX Connect system using 2 × SYBR Green Fast qRT-PCR Master Mix (Biotool) with a fast two-step amplification program. The relative expression was determined using the $2^{-\Delta\Delta Ct}$ method. The value obtained for each gene was normalized to that of the gene encoding β-actin. The primers utilized for qRT-PCR analysis of different ZDHHCs are listed in the Reagents and Tools Table.

## Mice and acute dissociation of DRG

TRPV1 knock-out mice ($Trpv1^{-/-}$) were generated by GemPharmatech (Gempharmatech Co., Ltd, Jiangsu, China) through CRISPR/Cas9-mediated genome editing. The dorsal root ganglion (DRG) of male mice were acutely dissociated, as described in a previous study (Tian et al, 2019), with minor modification. Briefly, 6 to 8-week-old wild-type C57BL/6 or $Trpv1^{-/-}$ mice were euthanized with $CO_2$ and then decapitated. The spinal cord was exposed and Lumbar 4–6 (L4–L6) DRGs in intervertebral foramen were dissociated, trimmed to remove the attached nerves and connecting tissues, and rinsed with ice-cold phosphate buffer saline (PBS). For recording TRPV1 currents in DRG neurons, L4–L6 DRG were acutely dissociated from mice, minced in cold Leibovitz's L-15 medium, and digested for 30–40 min at 37 °C in Hank's balanced salt solution supplemented with collagenase type IA (1 mg/ml), trypsin (0.4 mg/ml), and DNase I (0.1 mg/ml). After three washes with DMEM/F12 medium, the cells were gently dispersed, plated on glass coverslips coated with poly-D-lysine (0.5 mg/ml) and laminin (5 μg/ml), and cultured at 37 °C for an additional 2 h, allowing for cell attachment. DRG lysates for immunoblotting and immunoprecipitation experiments were prepared by homogenizing the acutely dissociated DRGs in ice-cold NP-40 lysis buffer (150 mM NaCl, 1 mM EDTA, 1% Nonidet P-40, 1% complete protease and phosphatase inhibitor cocktail, and 20 mM Tris–HCl; pH = 7.4).

## Cell culture and transfection

HEK293T and ND7/23 cell lines used in this study were obtained from the American Type Culture Collection (CRL-3216) and The European Collection of Authenticated Cell Cultures (92090903), authenticated by STR locus and tested negative for mycoplasma contamination. Cells were cultured in Dulbecco's Modified Eagle's Medium (DMEM) (Thermo Fisher Scientific, MA, USA) supplemented with 10% heat-inactivated fetal bovine serum (FBS, Gibco, Thermo Fisher Scientific) and 1% PS (50 units/ml penicillin + 50 μg/ml streptomycin), and maintained in standard cell culture condition (37 °C, 5% $CO_2$, and saturated humidity). Cells with ~80% confluence were transfected with the desired plasmids using either the standard calcium phosphate precipitation method or Lipofectamine 2000 (Invitrogen) following the manufacturer's instructions. For electrophysiological recordings, transfected HEK293T cells were seeded onto PLL (poly-L-lysine)-coated coverslips 4–6 h after transfection and cultured for an additional 12–24 h before the recording.

## Immunoprecipitation and immunoblotting

Immunoprecipitation was performed as previously described (Wang et al, 2018). Briefly, cells were harvested 24 h post-transfection with the NP-40 lysis buffer. Lysates were cleared of debris by centrifuging at 14,000 rpm for 10 min. The collected supernatant was incubated with indicated antibodies and Protein G agarose beads for 2 h, then the immunoprecipitants were washed with the modified NP-40 buffer containing 500 mM NaCl three times. Lastly, the cleaned immunoprecipitants were treated with SDS-based sample buffer and subjected to SDS-PAGE (4–20% or 12% gels) separation, followed by immunoblotting assay using standard protocol.

## Confocal imaging

HEK293T cells transfected with desired plasmids were fixed with 4% paraformaldehyde (Biosharp, BL539A) for 10 min at 4 °C, nuclei were counterstained with DAPI (Merck, D5492) for 15 min at room temperature in dark, and washed three times with phosphate-buffered saline (PBS). For BiFC, the GFP molecule is split into two fragments, neither of which fluoresces on its own (Hu et al, 2021). These fragments are then attached to different proteins. Only when these proteins interact with each other at the molecular level, the GFP fluorescence is restored. The stained cells were imaged on a Leica SP8 confocal microscope [×63 oil objective (NA 1.35)].

## Surface biotinylation assay

Surface biotinylation was performed following established protocols (Wang et al, 2018). Cells were firstly washed three times with ice-cold PBS supplemented with 1 mM $MgCl_2$ and 2.5 mM $CaCl_2$ (pH 8.0). Then Sulfo-NHS-LC-Biotin (0.25 mg/ml; Thermo Scientific, Waltham, MA, USA) was added and incubated with cells at 4 °C for 30 min with gentle rocking. The unbound biotin group was quenched by incubating the cells with 0.1 M glycine for 20 min at 4 °C. Biotin-labeled proteins were precipitated by incubating whole cell lysates with NeutrAvidin agarose beads (Thermo Fisher Scientific) overnight at 4 °C with gentle rocking. The beads were then washed three times with PBS (pH 8.0), and bound proteins were eluted with the boiling SDS-based sample buffer and subjected to immunoblotting analysis.

## Electrophysiology

Whole-cell patch-clamp recording was performed using a standard protocol. For transfected cells, the EGFP fluorescence was used to identify positively-transfected cells, and cells exhibiting green fluorescence were randomly selected for patch-clamp analysis. Currents were amplified using an Axopatch 200B amplifier (Molecular Devices, Sunnyvale, CA) and recorded by a BNC-2090/MIO acquisition system (National Instruments, Austin, TX) using QStudio developed by Dr. Feng Qin at State University of New York at Buffalo. Currents were typically sampled at 5 kHz and low-pass filtered at 1 kHz.

The standard bath solution for whole-cell recording contains (in mM): 140 NaCl, 5 KCl, 3 EGTA, and 10 HEPES (pH = 7.4, adjusted with NaOH); and the corresponding pipette solution contains (in mM): 150 CsCl, 5 EGTA, 10 HEPES (pH = 7.4, adjusted with CsOH). Pipettes were made from borosilicate glass capillaries (World Precision Instruments), fire-polished, and the pipette resistance was kept to 2–4 MΩ after filling pipette solution. Capsaicin (Cap) was dissolved in ethanol to make a stock solution (10 mM) and diluted with bath solution to desired concentrations before using. The acidic bath solution for evoking TRPV1 currents contains (in mM): 140 NaCl, 5 KCl, 5 EGTA, 1 $MgCl_2$, 10 glucose, and 10 MES, with pH adjusted to desired values by

NaOH. Cap or acid was applied using a gravity-driven local perfusion system. As determined by the conductance tests, the solution around a patch was fully controlled by the application of a solution with a flow rate of 100 µl/min or greater. All pharmacological experiments met this criterion. Unless otherwise indicated, all chemicals were from Sigma (Millipore Sigma, St. Louis, MO). All patch-clamp recordings were performed at room temperature (22–24 °C) except for experiments in Fig. 4G–L, where heat activation of TRPV1 was utilized.

## Ultrafast temperature jump achievement

A single emitter infrared laser diode (1550 nm) was designed to produce temperature jumps as previously described (Yao et al, 2009). The laser diode was driven by a pulsed quasi-CW current power supply (Stone Laser, Beijing, China), and the pulsing of the controller was controlled from a computer through BNC-2090/MIO data acquisition card, which was also responsible for patch-clamp recordings. The launched laser beam was transmitted by a multimode fiber with a core diameter of 100 µm. A blue laser line (460 nm) was coupled into the same fiber to aid alignment. Temperature was calibrated offline from the electrode current based on the temperature dependence of electrolyte conductivity. The temperature threshold for heat activation of TRPV1 was determined as the temperature at which the slow inward current was elicited.

## Behavioral test

Behavioral studies were performed with 6 to 8-week-old male C57BL/6 wild-type or $Trpv1^{-/-}$ mice. The mice of each genotype were randomly assigned to two groups, each receiving a different treatment: saline or carrageenan. The carrageenan-induced inflammation pain model was made by intraplantarly injecting the right hind paws of mice with 20 µL (2%, w/v) carrageenan, and the left hind paws with 20 µL vehicle (normal saline). Behavioral tests were conducted during the light phase of the light/dark cycle by a trained observer blind to the genotype and treatments. Mice were habituated to the testing room for 60 min prior to tests. The Hargreaves test was performed as previously described (Wang et al, 2018). The mouse was placed in a clear plexiglass cylinder on top of a temperature-controlled Plantar Test Instrument (Ugo Basile, Milan, Italy), which produces a high-intensity infrared light aimed at the plantar surface of the hindpaw. withdrawal latency of thermal hyperalgesia was determined by the onset of paw lifting, licking, and biting. A fixed infrared stimulus was used and the maximum stimulus time was set at 20 s to prevent tissue damage. For measuring mechanical allodynia, a dynamic plantar aesthesiometer (von Frey apparatus, Ugo Basile, Milan, Italy) were used (stimulus rate of 1 g/s; cutoff value of 10 g). Each mouse was placed individually in clear Plexiglas chambers ($8 \times 8 \times 12$ cm) and acclimated for 30 min before testing. The mechanical threshold was averaged from up to three stimuli. The capsaicin-induced nocifensive response was tested by injecting 10 µL capsaicin solution (0.2 µg/µL in saline) into the right hindpaw of the test mouse. The animal was placed in a plexiglass box. The total time spent on lifting, licking, and biting the injected paw was recorded during the first 10 min after injection.

## Acyl-biotin exchange (ABE) assay

The ABE assay was conducted according to the protocol described in the previous study (Brigidi and Bamji, 2013). Cells were lysed using NP-40 lysis buffer. The obtained lysate was incubated with 1 µg anti-TRPV1 antibody and 50 mM NEM (to alkylate the free cysteine residues) at 4 °C overnight. On the next day, the insoluble component was discarded by centrifugation at $13,500 \times g$ for 5 min. The cell lysate was precipitated by chloroform-methanol precipitation and washed twice using methanol to remove excess chemicals. Then the precipitate was dissolved in PBS containing 1% SDS, and treated with hydroxylamine buffer (0.7 M $NH_2OH$, 1 mM biotin, 0.2% Triton X-100, and 1% protease inhibitor) or Tris buffer (200 mM Tris, 1 mM biotin, 0.2% Triton X-100, and 1% protease inhibitor) in dark for 1 h. After chloroform-methanol precipitation, the protein was resolubilized in PBS containing 1% Triton X-100 and 0.2% SDS. TRPV1 proteins were immunoprecipitated with Protein G agarose beads and subjected to immunoblotting analysis, with HRP-streptavidin and anti-TRPV1 antibody being used to detect the palmitoylated TRPV1 and total TRPV1, respectively.

## AAV virus and intrathecal injection

AAV virions were produced by Brain Case Co. Ltd. (Shenzhen, China). Briefly, the shRNA of Scramble, mZDHHC4, and mAPT1 were PCR-amplified individually and cloned using standard methods into pAAV-U6-mcs-CMV-EGFP vector. AAV-shScramble ($5.76 \times 10^{12}$ vg/ml), AAV-shZDHHC4 ($5.60 \times 10^{12}$ vg/ml) and AAV-shAPT1 ($5.36 \times 10^{12}$ vg/ml) were produced by transfecting the desired pAAV plasmid into HEK293 cells. 72 h after transfection, viral particles were collected by phosphate-buffered saline (PBS) and purified by the iodixanol step-gradient ultracentrifugation method. Intrathecal injection of AAV was performed as described previously (Li et al, 2019). Briefly, mice were secured on the operating board, and the injection site was dehaired. The intervertebral space between L5–L6 was palpated manually, and 8 µL of AAV virus solution was injected intrathecally at this site using a 27G needle.

## Data analysis

Densitometry was performed using ImageJ software (National Institutes of Health, NIH) to quantitatively analyze the bands on images of western blots. Electrophysiological data were analyzed offline with Qstudio developed by Dr. Feng Qin at State University of New York at Buffalo, Clampfit (Molecular Devices, Sunnyvale, CA), IGOR (Wavemetrics, Lake Oswego, OR, USA), SigmaPlot (SPSS Science, Chicago, IL, USA), and OriginPro (OriginLab Corporation, MA, USA). For concentration-response analysis, the modified Hill equation was used: $Y = A_1 + (A_2 - A_1)/[1 + 10$ ^ $(logEC_{50} - X) * n_H]$, in which $EC_{50}$ is the half maximal effective concentration, and $n_H$ is the Hill coefficient. Data are presented as mean ± standard deviation (SD), with statistical significance assessed using unpaired Student's *t*-test for two-group comparisons, or one-way analysis of variance (ANOVA) with post hoc Dunnett/Turkey analysis for comparisons involving three or more groups. The *n* value represents the number of biological replicates. Significant difference is accepted at $p < 0.05$.

## Study approval

All animals were housed in a pathogen-free animal facility at Wuhan University and all animal experiments were in accordance with protocols approved by the Institutional Animal Care and Use Committee of Wuhan University (No. WDSKY0201804) and

adhered to the Chinese National Laboratory Animal-Guideline for Ethical Review of Animal Welfare. The animals were euthanized with $CO_2$, followed by various studies.

## Data availability

All data presented in the manuscript are publicly accessible at https://www.ebi.ac.uk/biostudies/studies/S-BSST1442 with the BioStudies accession number S-BSST1442 and https://doi.org/10.6019/S-BSST1442.

The source data of this paper are collected in the following database record: biostudies:S-SCDT-10_1038-S44319-024-00317-0.

## Peer review information

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

## Acknowledgements

We extend our gratitude to all members of the Yao lab for their insightful comments and valuable discussions on this study. Part of this work was performed at the core facilities platform at the College of Life Sciences, Wuhan University, where technical support was provided. This work was supported by grants from the National Key Research and Development Program of China (2023YFF1204000) and National Natural Science Foundation of China (32171147, 32371200, 32370761, and 31929003).

## Author contributions

**Youjing Zhang**: Data curation; Formal analysis; Validation; Methodology; Writing—original draft; Writing—review and editing. **Mengyu Zhang**: Data curation; Formal analysis; Validation; Methodology; Writing—original draft. **Cheng Tang**: Data curation; Formal analysis; Validation; Investigation; Methodology; Writing—original draft; Writing—review and editing. **Junyan Hu**: Data curation; Formal analysis; Investigation. **Xufeng Cheng**: Data curation; Formal analysis; Investigation. **Yang Li**: Data curation; Formal analysis; Investigation. **Zefeng Chen**: Data curation; Formal analysis; Investigation. **Yuan Yin**: Data curation; Formal analysis. **Chang Xie**: Resources; Data curation; Formal analysis; Funding acquisition; Validation; Investigation; Methodology. **Dongdong Li**: Conceptualization; Formal analysis; Supervision; Validation; Investigation; Visualization; Methodology; Writing—original draft; Writing—review and editing. **Jing Yao**: Conceptualization; Resources; Data curation; Software; Formal analysis; Supervision; Funding acquisition; Validation; Investigation; Visualization; Methodology; Writing—original draft; Project administration; Writing—review and editing.

Source data underlying figure panels in this paper may have individual authorship assigned. Where available, figure panel/source data authorship is listed in the following database record: biostudies:S-SCDT-10_1038-S44319-024-00317-0.

## Disclosure and competing interests statement

The authors declare no competing interests.

