## [Peer Review File · EMBO Reports]

Palmitoylation by ZDHHC4 inhibits TRPV1-mediated nociception

Youjing Zhang, Mengyu Zhang, Cheng Tang, Junyan Hu, Xufeng Cheng, Yang Li, Zefeng Chen, Yuan Yin, Chang Xie, Dongdong, and Jing Yao

Corresponding author(s): Jing Yao (jyao@whu.edu.cn) , Dongdong Li (dongdong.li@inserm.fr)

Review Timeline:

Transfer Date:	18th Sep 24
Editorial Decision:	1st Oct 24
Revision Received:	16th Oct 24
Editorial Decision:	22nd Oct 24
Revision Received:	24th Oct 24
Accepted:	30th Oct 24

Editor: Achim Breiling

Transaction Report: A revised version of this manuscript was transferred to EMBO reports following peer review at The EMBO Journal.

Referee #2:

The authors performed a substantial amount of experiments, which improved some aspects of the manuscript. Unfortunately, I cannot recommend acceptance, as the manuscript still suffers from significant problems, which are outlined below:

1. Most importantly, the new behavioral data did not make the claim conveyed in the title "TRPV1 palmitoylation mediated by ZDHHC4 underlies inflammatory pain relief" any more convincing. Looking at the tiny difference in Fig 8D2, even if significant statistically in the AUC plot, one could argue that palmitoylation has a very modest contribution to recovery from inflammatory pain. The authors argue in the manuscript, that the fast recovery from thermal hyperalgesia may have obscured the effect of ZDHHC4 knockdown. I think this not a very good argument: if ZDHHC4 plays an important role, would not you think recovery is fast because of ZDHHC4? So if you knock it down would not you expect a larger effect on recovery because you baseline is fast ?

I think the authors should take out the reference to inflammatory pain relief from the title, and restrict the statement to palmitoylation inhibiting TRPV1 by lysosomal degradation. Also the data in Figure 8D1 D2 should be replotted in a way that the individual curves are visible, as on the current version it very hard to see what is what. One option is to plot the saline injected ones separately. A related point, the authors claim in line 127 that the mechanical allodynia lasted for almost a week. This is impossible to see from the plots as they are now. After replotting, the authors, my (or may not) need to reevaluate this statement.

2. The difference is slightly more convincing in the mechanical hyperalgesia, but the role of TRPV1 in mechanical sensitivity is less clear than in thermal hyperalgesia. The authors cite two papers in their response (but not in the MS) that support the role of TRPV1, but there are also plenty of papers that claim TRPV1 is not playing a role in mechanical hypersensitivity, for example: Caterina et al Science 2000, and Davis et al Nature 2000.

3. I am also puzzled by the way the authors handled the question about the JAK-STAT pathway. They performed experiments to support their hypothesis, and they show them (with n=1 ?) in the response, but not in the manuscript. I would either take the JAK-STAT data out, or put in the additional experiments with proper statistics.

4. Figure 1D2, the increase in palmitoylation is a key measurement, it needs a significance test.

5. Figure 3A and 3C needs statistical summary and a significance test, given that the authors claim it is a significant effect.

6. Line 360 this was significantly reversed by APT1 knockdown (Supplementary Figure S4C1 and C2). - there is no significance test provided for this figure

Minor comments:

7. Line 216 TRPV1 abundance was remarkably reduced by the overexpressed ZDHHC4 (Figure 3B). This is a tiny effect, even if significant, I would not call it remarkable.

7. The authors show a much larger effect on TRPV1 current amplitudes than on surface expression. Is it possible that lysosomal degradation is not the sole mechanism of TRPV1 inhibition? The authors may consider commenting on this.

9. Lines 403 - 406: The discussion on capsaicin as an analgesic is quite problematic. Capsaicin as an analgesic has been considered as a topical modality, because it desensitizes TRPV1 positive nerve fibers, but it does so only after evoking severe pain first. This limits its usage and patient compliance. I do not think capsaicin has ever been seriously considered as a systemic analgesic.

10: Line 939 fluorescence intensity and proportion of interacting regions were analyzed. Unclear fluorescence of what, and what is meant on "proportion of interacting regions"

Referee #3:

The authors have diligently responded to my critique, point-by-point, and now present a well-rounded manuscript, reporting interesting and novel findings, as indicated in my prior review, with previous shortcomings satisfactorily addressed.

RE: EMBOJ-2024-116804R entitled "TRPV1 palmitoylation mediated by ZDHHC4 underlies inflammatory pain relief"

General reply: We thank sincerely the editor and the reviewers for their positive assessment and constructive comments on our manuscript. Following the suggestions raised by the referee #2, we have clarified certain statements, and improved the data presentation. Our point-to-point responses are described below, with corresponding changes tracked in the manuscript.

Referee #2:

The authors performed a substantial amount of experiments, which improved some aspects of the manuscript. Unfortunately, I cannot recommend acceptance, as the manuscript still suffers from significant problems, which are outlined below:

1. Most importantly, the new behavioral data did not make the claim conveyed in the title "TRPV1 palmitoylation mediated by ZDHHC4 underlies inflammatory pain relief" any more convincing. Looking at the tiny difference in Fig 8D2, even if significant statistically in the AUC plot, one could argue that palmitoylation has a very modest contribution to recovery from inflammatory pain. The authors argue in the manuscript, that the fast recovery from thermal hyperalgesia may have obscured the effect of ZDHHC4 knockdown. I think this not a very good argument: if ZDHHC4 plays an important role, would not you think recovery is fast because of ZDHHC4? So if you knock it down would not you expect a larger effect on recovery because you baseline is fast?

I think the authors should take out the reference to inflammatory pain relief from the title, and restrict the statement to palmitoylation inhibiting TRPV1 by lysosomal degradation. Also the data in Figure 8D1 D2 should be replotted in a way that the individual curves are visible, as on the current version it very hard to see what is what. One option is to plot the saline injected ones separately. A related point, the authors claim in line 127 that the mechanical allodynia lasted for almost a week. This is impossible to see from the plots as they are now. After replotting, the authors, my (or may not) need to reevaluate this statement.

Reply: Thank you for these insightful comments. We acknowledge that in the carrageenan-induced inflammatory model, manipulating TRPV1 palmitoylation by knocking down ZDHHC4 or APT1 partially affected thermal hyperalgesia. This suggests other parallel mechanisms contributing to the pain resolution. For instance, a recent paper (Defaye et al, 2024) suggests that interferon-stimulated genes modulate pain resolution, while involving TRPV1 downregulation. However, the mechanism underlying TRPV1 downregulation has remained unresolved. Our study here provides

evidence that ZDHHC4-mediated palmitoylation contributes to TRPV1 degradation during inflammatory pain resolution. We now include this information in the discussion (Page 22, Lines 428-439. Following the reviewer's suggestion, we have also removed the ambiguous argument (page 18, lines 360-361 '*This may have obscured the effects of ZDHHC4 and APT1 knockdown on thermal hyperalgesia resolution.*').

As for the title, we now revise to "Palmitoylation mediated by ZDHHC4 inhibits TRPV1 nociception" to appropriately position the main findings of our study

To clarify Figure 8D₁-D₂, we have now separated the saline data to **Supplementary Figure 4D₁-D₂**. The clarity of the main figure (Fig 8D₁-D₂) is indeed improved now.

Additionally, we have removed the statement "lasting almost a week" (Page 7, Line 124).

2. The difference is slightly more convincing in the mechanical hyperalgesia, but the role of TRPV1 in mechanical sensitivity is less clear than in thermal hyperalgesia. The authors cite two papers in their response (but not in the MS) that support the role of TRPV1, but there are also plenty of papers that claim TRPV1 is not playing a role in mechanical hypersensitivity, for example: Caterina et al Science 2000, and Davis et al Nature 2000.

Reply: We agree that the role of TRPV1 mechanical sensitivity has been in deliberation. We show here the observation on the mechanical hyperalgesia, echoing the implication of TRPV1 in mechanical sensitivity as suggested by other studies as mentioned in the manuscript. Nonetheless, we thank the reviewer to point the necessity to also incorporate the studies arguing against the role of TRPV1 in mechanical hypersensitivity. We have now mentioned this point in the manuscript (Page 22, Lines 435-441).

3. I am also puzzled by the way the authors handled the question about the JAK-STAT pathway. They performed experiments to support their hypothesis, and they show them (with n=1?) in the response, but not in the manuscript. I would either take the JAK-STAT data out, or put in the additional experiments with proper statistics.

Reply: We apologize for the ambiguity. We did conduct several repeats ($n = 3$) but only presented one typical blot in the last response letter. In this revision, we have added appropriate statistical analysis for this figure and incorporated the data into the revised **Supplementary Figure 2C₁-C₂**.

4. Figure 1D₂, the increase in palmitoylation is a key measurement, it needs a

significance test.

Reply: We have now added statistical analysis for this figure (**revised Figure 1D₂**). The result reveals the significance.

5. Figure 3A and 3C needs statistical summary and a significance test, given that the authors claim it is a significant effect.

Reply: The statistical analysis for this figure (revised Figure 3A and 3C) is now displayed, showing statistical significance.

6. Line 360 this was significantly reversed by APT1 knockdown (Supplementary Figure S4C1 and C2). - there is no significance test provided for this figure

Reply: Again, we have now added statistical analysis and significance test for this figure (revised Supplementary Figure 4C₂), which shows APT1 knockdown significantly accelerated TRPV1 degradation at day 2, 4 and 7 in comparison to the ZDHHC4 knockdown group.

Minor comments:

7. Line 216 TRPV1 abundance was remarkably reduced by the overexpressed ZDHHC4 (Figure 3B). This is a tiny effect, even if significant, I would not call it remarkable.

Reply: The word 'remarkably' is now removed (Page 11, Line 213).

8. The authors show a much larger effect on TRPV1 current amplitudes than on surface expression. Is it possible that lysosomal degradation is not the sole mechanism of TRPV1 inhibition? The authors may consider commenting on this.

Reply: A quantitative comparison between biochemical and electrophysiological approaches may not be so straightforward. Western blot analysis is a semiquantitative method for determining protein abundance, whereas electrophysiological recording assesses the functional expression of ion channels. It is possible that not all channels located on the membrane are functional. That said, we fully agree with the reviewer that lysosomal degradation represents one of the multiple mechanisms responsible for TRPV1 inhibition. For instance, we have shown that the activity-gated recycling of TRPV1 channels can also shape their functional response (Tian *et al*, 2019).

9. Lines 403 - 406: The discussion on capsaicin as an analgesic is quite problematic. Capsaicin as an analgesic has been considered as a topical modality, because it

desensitizes TRPV1 positive nerve fibers, but it does so only after evoking severe pain first. This limits its usage and patient compliance. I do not think capsaicin has ever been seriously considered as a systemic analgesic.

Reply: We have now removed these statements.

10: Line 939 fluorescence intensity and proportion of interacting regions were analyzed. Unclear fluorescence of what, and what is meant on "proportion of interacting regions"

Reply: We apologize for not making this clear. The bars on the left panel illustrate the quantification of fluorescence intensity of interaction area in the DRGs of mice that received either saline or carrageenan injections. The bars on the right panel quantify the size for colocalization area of TRPV1 and ZDHHC4 in the DRGs of saline- or carrageenan-injected mice. We have revised this information (end of figure 2 legend) to improve clarity.

Referee #3:

The authors have diligently responded to my critique, point-by-point, and now present a well-rounded manuscript, reporting interesting and novel findings, as indicated in my prior review, with previous shortcomings satisfactorily addressed.

Reply: Many thanks for your favorable endorsement.

References:

Defaye M, Bradaia A, Abdullah NS, Agosti F, Iftinca M, Delanne-Cuménal M, Soubeyre V, Svendsen K, Gill G, Ozmaeian A *et al* (2024) Induction of antiviral interferon-stimulated genes by neuronal STING promotes the resolution of pain in mice. *Journal of Clinical Investigation* 134

Tian Q, Hu J, Xie C, Mei K, Pham C, Mo X, Hepp R, Soares S, Nothias F, Wang Y *et al* (2019) Recovery from tachyphylaxis of TRPV1 coincides with recycling to the surface membrane. *Proc Natl Acad Sci U S A* 116: 5170-5175

Dear Prof. Yao,

Thank you for the transfer of your further revised manuscript to EMBO reports. Going through your p-b-p-response, and after discussing the study with the former handling editor at The EMBO Journal, I consider the remaining referee points as adequately addressed.

Before I can proceed with formal acceptance, I have these editorial requests I ask you to address in a final revised manuscript:

- Please upload a completed author checklist, which you can download from our author guidelines (<https://www.embopress.org/page/journal/14693178/authorguide>). Please insert page numbers in the checklist to indicate where the requested information can be found in the manuscript. The completed author checklist will also be part of the peer review file.

- We now use CRediT to specify the contributions of each author in the journal submission system. CRediT replaces the author contribution section. Please use the free text box to provide more detailed descriptions and do NOT provide your final manuscript text file with an author contributions section. See also our guide to authors: <https://www.embopress.org/page/journal/14693178/authorguide#authorshipguidelines>

- Please order the manuscript sections like this, using these names:
Title page - Abstract - Keywords - Introduction - Results - Discussion - Methods - Data availability section - Acknowledgements - Disclosure and Competing Interests Statement - References - Figure legends

- Please make sure that the number "n" for how many independent experiments were performed, their nature (biological versus technical replicates), the bars and error bars (e.g. SEM, SD) and the test used to calculate p-values is indicated in the respective figure legends. Please also check that all the p-values are explained in the legend, and that these fit to those shown in the figure. Please provide statistical testing where applicable. Please avoid the phrase 'independent experiment', but clearly state if these were biological or technical replicates. Please also indicate (e.g. with n.s.) if testing was performed, but the differences are not significant. In case n=2, please show the data as separate datapoints without error bars and statistics. See also: <http://www.embopress.org/page/journal/14693178/authorguide#statisticalanalysis>

If n<5, please show single datapoints for diagrams. Moreover:

- Please note that information related to n is missing in the legends of figures 1a1-a2, b2, c2, d2 ; 2e2; 3b, d, f; 4c; 5c-d; 6d, f2; 7c; 8c-d2; supplementary figures 2b; 4c2.

- Although 'n' is provided, please describe the nature of entity for 'n' in the legends of figures 4b, e-f, h; 5f-g; 7f-g.

- Please note that the error bars are not defined in the legends of figures 1a1-a2, b2, c2, d2; 2e2, g; 3b, d, f; 4b-c, e-f, h-i; k-l; 5c-d, f-g; 6d, f2; 7c, f-g; 8c-e2; supplementary figures 2b; 3c-d; 4c2.

- Please define the annotated p values ***/**/* as well as provide the exact p-values for the same in the legend of figure 4c, f; 5f; 7c, f; 8c-d2, e2; as appropriate.

- Please note that the exact p values are not provided in the legends of figures 2a; 3b, d, f; 5c-d; 6d.

- Please indicate the statistical test used for data analysis in the legends of figures 4c, f, l; 5f; 7c, f; 8c-e2.

- Please note that in figure 2a; there is a mismatch between the annotated p values in the figure legend and the annotated p values in the figure file that should be corrected.

- Please note that the yellow lines are not defined in the legend of figure 2f. This needs to be rectified.

- Please do NOT use panel labels like C1, C2 ... but just alphabetic labels (in all figures, including the Appendix figures). There are several panels labeled like this. Please change.

- Please add scale bars of similar style and thickness to all microscopic images (main, EV and Appendix figures), using clearly visible black or white bars (depending on the background). Please place these in the lower right corner of the images themselves. Please do not write on or near the bars in the image but define the size in the respective figure legend. Presently, some scale bars are too thin. Please check.

- Please name the Appendix file 'Appendix' and provide it as pdf. Please add a table of content and with page numbers to the title page and remove the author names and affiliation information. It is sufficient to state 'Appendix for ...' followed by the title of the paper. Please use the nomenclature Appendix Figure Sx for the Appendix figures and for their callouts.

- Please move the table in the methods section and Table S1 to the reagents & tools table (see below).

- All Materials and Methods need to be described in the main text using our 'Structured Methods' format, which is required for all research articles. According to this format, the Materials and Methods section should include a Reagents and Tools Table (listing key reagents, experimental models, software, and relevant equipment and including their sources and relevant identifiers), uploaded as separate file, followed by a Methods section in which we encourage the authors to describe their

methods using a step-by-step protocol format with bullet points, to facilitate the adoption of the methodologies across labs. More information on how to adhere to this format as well as downloadable templates (.doc) for the Reagents and Tools Table can be found in our author guidelines (section 'Structured Methods'):

- Thank you for providing the source data. Please provide all source data for one figure in a single folder, zipped and then uploaded. For EV and/or Appendix figures, all source data can be combined/zipped together into one folder and uploaded. Presently, the folder for Fig. 2 uploaded can't be opened (unZIPed). Please check.

In addition, I would need from you uploaded separately:

Best,

All editorial and formatting issues were resolved by the authors.

Dear Prof. Yao,

Thank you for the submission of your further revised manuscript to EMBO reports. I now looked through this and identified some remaining editorial points that I ask you to address in a final revised manuscript:

- Please have your manuscript carefully proofread by a native speaker. There are still grammatical errors present (e.g. right the first word of the abstract needs to be removed - it should be 'Transient receptor potential vanilloid 1 (TRPV1) is a capsaicin-sensitive ion channel implicated in pain sensation.'). Please also revise the abstract, using a more scientific language, in particular please explain what is meant with 'cooling-down function'.

- We request that the error bars and the nature of 'n' is defined/described in each legend (and not only somewhere in the methods section), as is the statistical test used. Thus, please add to each legend (main and Appendix figures) a 'Data Information' section explaining the statistics used and providing information regarding replicates and scales. See:

- You also indicate n=3 for panels showing Western blots. What does this mean? One representative blot of three replicate experiments is shown? If yes, please add this information.

- Please also indicate in diagrams (with n.s.) or in the legend if testing was performed, but the differences were not significant.

- The writing in the synopsis image provided is too small when sized to the correct format (the exact width of 550 pixels and a height of not more than 400 pixels). See attached.

Please provide an image (with the correct size) with bigger fonts.

All editorial and formatting issues were resolved by the authors.

Prof. Jing Yao
College of Life Sciences at Wuhan University
Department of Cell Biology
299 Bayi Road
Wuhan, Hubei 430072
China

Dear Prof. Yao,

I am very pleased to accept your manuscript for publication in the next available issue of EMBO reports. Thank you for your contribution to our journal.

Yours sincerely,
